# Initial state of DNA-Dye complex sets the stage for protein induced fluorescence modulation

Fahad Rashid[1,2], Vlad-Stefan Raducanu[1,2], Manal S. Zaher [1,2], Muhammad Tehseen[1], Satoshi Habuchi [1] & Samir M. Hamdan [1]

Protein-induced fluorescence enhancement (PIFE) is a popular tool for characterizing protein-DNA interactions. PIFE has been explained by an increase in local viscosity due to the presence of the protein residues. This explanation, however, denies the opposite effect of fluorescence quenching. This work offers a perspective for understanding PIFE mechanism and reports the observation of a phenomenon that we name protein-induced fluorescence quenching (PIFQ), which exhibits an opposite effect to PIFE. A detailed characterization of these two fluorescence modulations reveals that the initial fluorescence state of the labeled mediator (DNA) determines whether this mediator-conjugated dye undergoes PIFE or PIFQ upon protein binding. This key role of the mediator DNA provides a protocol for the experimental design to obtain either PIFQ or PIFE, on-demand. This makes the arbitrary nature of the current experimental design obsolete, allowing for proper integration of both PIFE and PIFQ with existing bulk and single-molecule fluorescence techniques.

---

[1] King Abdullah University of Science and Technology, Division of Biological and Environmental Sciences and Engineering, Thuwal 23955, Saudi Arabia. [2] These authors contributed equally: Fahad Rashid, Vlad-Stefan Raducanu, Manal S. Zaher. Correspondence and requests for materials should be addressed to S.M.H. (email: samir.hamdan@kaust.edu.sa)

In recent years, protein-induced fluorescence enhancement (PIFE) has gained popularity as a stand-alone or complementary single-molecule (SM) imaging assay. This popularity mainly stems from its simple labeling requirements involving the use of a single fluorophore[1] and its ability to measure changes in a distance-dependent manner that supersedes the Förster resonance energy transfer (FRET) range[2]. As a photophysical phenomenon, PIFE occurs in environmentally sensitive fluorophores of the cyanine family (Supplementary Fig. 1). PIFE experiments with cyanine dyes coupled to DNA or to protein illustrate that neither the dye coupling nor the presence of the interacting molecule(s) induce a bathochromic or hypsochromic shift[3]. Therefore, the explanation of the environmental sensitivity of cyanine dyes is ascribed to the cis-trans photoisomerization of their two indole cyclic groups around the polymethine bond (Fig. 1b and Supplementary Fig. 1)[3,4]. Even though both cis and trans isomers can exist in the ground state, it is the trans isomer that is mostly excited since it has a larger absorption cross-section, reflecting its higher symmetry and larger absorption transition dipole moment[3,5–9].

Upon excitation, only the trans $S_1^*$ to trans $S_0$ transition results in fluorescence emission, with photoisomerization as one of its competitors. It is well-accepted that the isomerization in the excited state goes through a metastable 90°-twisted intermediate state during the rotation, which rapidly and non-radiatively de-excites to the ground state (Fig. 1a)[3–6,10,11]. Within this paradigm, PIFE has been explained by an analogy to an increase in the local viscosity due to the presence of the protein residues decreasing the rate of photoisomerization from trans $S_1^*$ to twisted $S_1^*$ (Fig. 1a). The dye's local rotational freedom has been suggested to be directly related to its ability to photoisomerize[12–16].

Existing models rely on the idea that proteins influence the rate of photoisomerization from trans $S_1^*$ to twisted $S_1^*$ through the concepts of steric hindrance/restriction[17] and specific contact with certain residues[3,4]. However, these models predict only fluorescence enhancement and deny the existence of an opposite phenomenon given that the presence of the protein can only increase the local viscosity. If the dye is in contact with the same protein residues, these models assume that the change in fluorescence will not depend on the initial fluorescence of the DNA-Dye (structure and sequence of DNA). Collectively, these observations suggest that the protein is primarily responsible for the fluorescence enhancement in PIFE and the role of the DNA, itself, is trivial.

To expand the current models of PIFE, one would have to consider the mediator that propagates the interactions and to which the dye is attached. Coupling fluorophores to DNA or to a protein, via at least one linker, has been shown to enhance their fluorescence, mostly by a partial rigidization of the

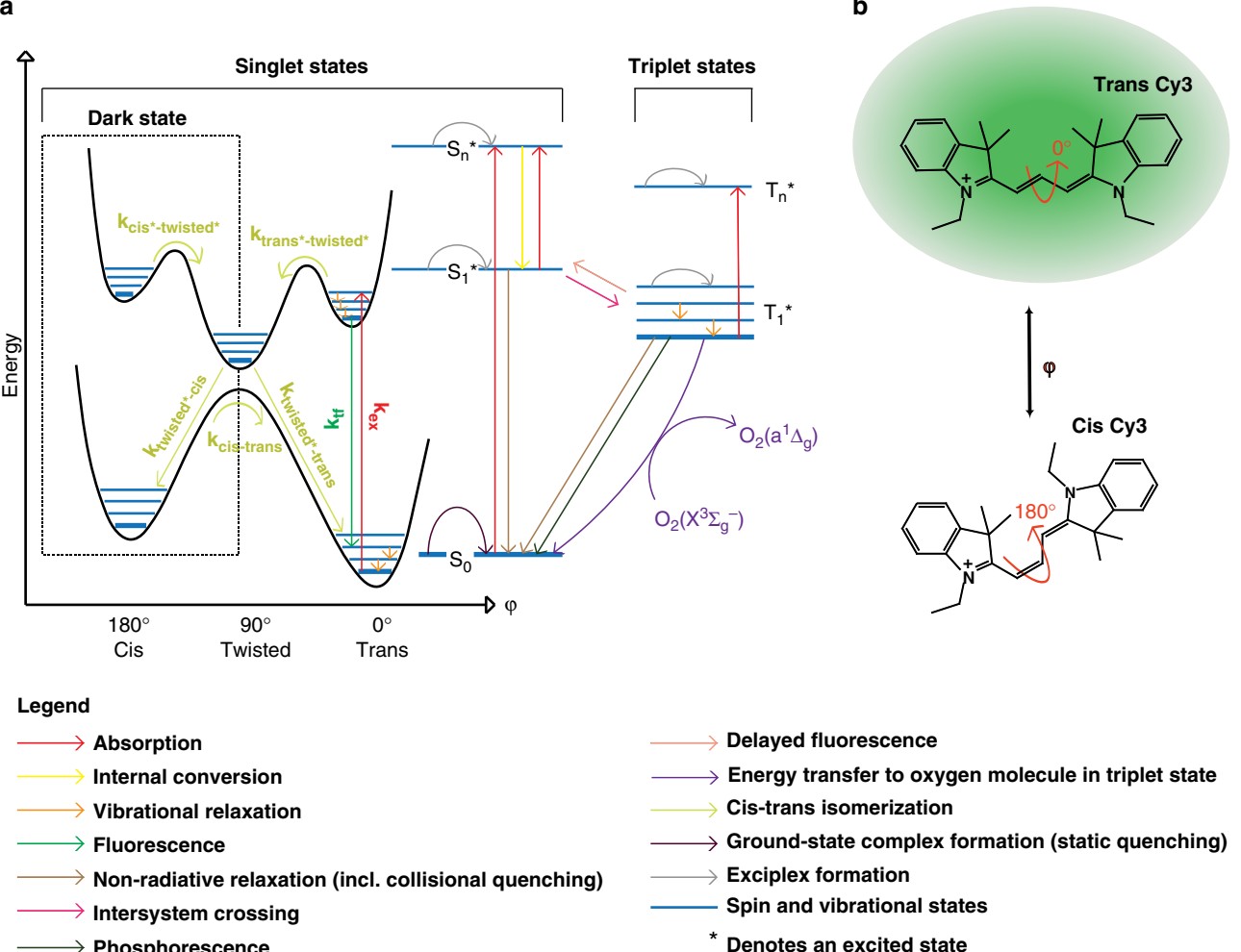

**Fig. 1** Energy landscape of a cyanine dye. **a** Schematic representation of potential energy landscape for $S_0$ (ground singlet) and $S_1^*$ (first excited singlet) states as a function of single dihedral angle of the Cy3 polymethine bond, superposed on a Jablonski diagram. **b** Schematic representation of the Cy3 dye molecule and one of its dihedral angles, in fluorescent trans and dark cis configurations

fluorophore[6,8,10,14,18–20]. These studies also defined some of the parameters contributing to the fluorescence properties of DNA-coupled Cy-dyes, including DNA sequence, dye position, and the overall DNA structure. With such parameters in mind, we need to consider an extra dimension in PIFE experiments, because the DNA itself can restrict the fluorophore's photoisomerization through various interactions, therefore dictating the initial state of fluorescence. However, current characterizations of PIFE and DNA-coupled Cy-dyes neither offer a systematic or a quantitative correlation between the initial fluorescence of the DNA-Dye system and the final fluorescence after binding of the protein, nor do they provide detailed guidelines on how to design and control PIFE.

To characterize the relationship between the initial and final fluorescence states of DNA-coupled Cy-dyes upon protein interaction, we investigated a phenomenon that we call protein-induced fluorescence quenching (PIFQ), because it showed the opposite effect of PIFE upon binding of the proteins to certain DNA–Dye complexes. It is worth mentioning that none of the known fluorescence quenching mechanisms[21–23] can explain the PIFQ effect observed here. Therefore, we present a different perspective on PIFE and PIFQ by investigating the change in fluorescence induced by disrupting the initial DNA-Dye structure upon protein binding. From this perspective, the initial DNA-Dye structure is the determinant factor for the initial state of fluorescence, which in turn sets the stage for the fluorophore to experience either enhancement or quenching in the presence of an external modulator (e.g., bound proteins, annealed complementary DNA strands, and specific DNA structures induced by metal ions). However, the final state of fluorescence is dictated by the structure and interactions of the dye within the final complex. Our perspective does not exclude the possibility of steric hindrance/restriction and/or specific contact with certain residues due to the presence of the protein contributing to the final state, but rather adds one additional variable, that is, the initial state of fluorescence of the DNA-Dye system. Furthermore, similar to the concept of nucleic acid-induced fluorescence enhancement (NAIFE)[12,24], we introduce nucleic acid-induced fluorescence quenching (NAIFQ) as an opposite effect. Additionally, we show that similarly to PIFE, PIFQ is suitable in both ensemble and single-molecule assays.

## Results

**DNA-mediated bidirectional fluorescence modulation**. We recently used smFRET to characterize the mechanism of double flap (DF) substrate recognition by DNA replication and repair Flap Endonuclease 1 (FEN1)[25]. Briefly, FEN1 cleaves excess 5′-flaps from the DNA by threading the 5′-flap into a capped helical gateway[25,26]. In-vitro, DF is made by annealing three different oligonucleotides to create a nick bearing a 5′-flap of single-stranded DNA (ssDNA) of variable lengths and 1 nucleotide (nt) 3′-flap; these substrates are named DF-(length of 5′-flap), (length of 3′-flap). Previously, we used a labeling scheme that employs Cy3 attached, through phosphoramidite linkage (referred to, from here onwards, as pCy3), at the tip of the 5′-flap (Fig. 2a)[25,27,28]. We were puzzled by a photophysical protein-induced pCy3 quenching[25]. Steady-state fluorescence spectra of pCy3 in DF-6,1 exhibited major FEN1-induced fluorescence quenching without causing any spectral shift (Supplementary Fig. 2a). This quenching showed flap-length dependence (Supplementary Fig. 2b), similar to PIFE distance-dependence[2]. Known quenching mechanisms cannot explain this observed effect, since FEN1 does not contain any iron sulfur cluster and carbocyanine dyes cannot be quenched, via PET, by tryptophan residues or guanosine nucleobases[29,30]. We thus conclude that this pCy3 quenching

is a distinct observation and refer to it as protein-induced fluorescence quenching (PIFQ), analogous to PIFE.

Next, we used FEN1/DF system to characterize the nature of PIFQ and the factors that affect it. We investigated whether the observed PIFQ was mediated by DNA itself or by FEN1 directly affecting the photophysics of pCy3. We measured time-resolved fluorescence lifetime of pCy3, for each flap length, in three contexts: (1) 5′-pCy3-labeled oligo, pCy3-labeled DF substrate without (2) and with FEN1 (3) (Fig. 2a). This confirmed FEN1-induced quenching behavior (e.g., DF-6,1 case; Fig. 2b). Surprisingly, the lifetime of pCy3 in the oligo alone was significantly lower than that of DF. Hence, upon oligo annealing to form DF, the fluorescence of pCy3 was dramatically enhanced and FEN1 binding reduced this enhancement back to a similar level as in the oligo alone. This fluorescence enhancement in the context of nucleic acid–nucleic acid interactions, such as upon DF formation, is referred to as nucleic acid-induced fluorescence enhancement (NAIFE)[12]. The same trend was observed across the different flap lengths (Fig. 2c). Interestingly, although the different oligos had similar initial fluorescence lifetimes, their NAIFE levels decreased as flap length increased. This suggests that the observed NAIFE is primarily caused by the interaction of Cy3 with the hybrid DNA. The percentage of lifetimes change vs. flap length shows an anti-correlated behavior, characterized by an increase in fluorescence upon formation of the substrate that is counteracted by FEN1 binding to the DF substrates (Fig. 2d). Therefore, this NAIFE is driving the subsequent quenching effect. Thus, the observation can be better understood from the perspective of the DNA as a mediator of fluorescence change rather than the protein being the active re-configurator of fluorescence.

To account for NAIFE upon formation of the DF substrate and the subsequent PIFQ, we considered various possibilities. We first ruled out any ground state-induced static quenching by showing that FEN1 binding does not cause any significant change in the absorbance of DF substrates (Supplementary Fig. 2c), and that fluorescence lifetime and quantum yield (Fig. 2c, d and Supplementary Fig. 2e, f) follow a similar trend. To probe whether the fluorescence modulations in FEN1/DF system are analogous to those in PIFE, particularly with respect to the modulation of photoisomerization, we exchanged pCy3 with Cy3B in our three-context system for three flap lengths. Cy3B is an analog of Cy3 dye, but with a rigid inter-heterocyclic construction that renders it incapable of photoisomerization[2,3] (Supplementary Fig. 1). The fluorescence lifetimes of Cy3B-DNA in the three-context system across different flap lengths showed no difference (Fig. 2e). Moreover, the theoretical total non-radiative excited-state lifetime loss anti-correlated with the fluorescence lifetime of pCy3 in the three-context system (Supplementary Fig. 2d), and was significantly higher than that of Cy3B, accounting for a photoisomerization loss. The photo-isomerization loss was confirmed by estimating directly the photoisomerization rate as described in SI Methods (Supplementary Fig. 2d-inset). In conclusion, the Cy3B experiments, along with the theoretical total non-radiative lifetime loss and photoisomerization rates, show that for the pCy3-labeled system, both NAIFE and PIFQ stem from the excited state cis-trans photoisomerization.

To probe the formation of DNA–Dye interactions[8,31–33], we performed steady-state and time-resolved anisotropy measurements with pCy3-labeled DF in the three-context system; as a control, we performed these measurements with Cy3B-labeled DF (Supplementary Fig. 2g, h). The results showed that there are interactions restricting the dye's rotational freedom and these interactions increase upon substrate annealing and decrease upon FEN1 binding. Moreover, there is a strong correlation between

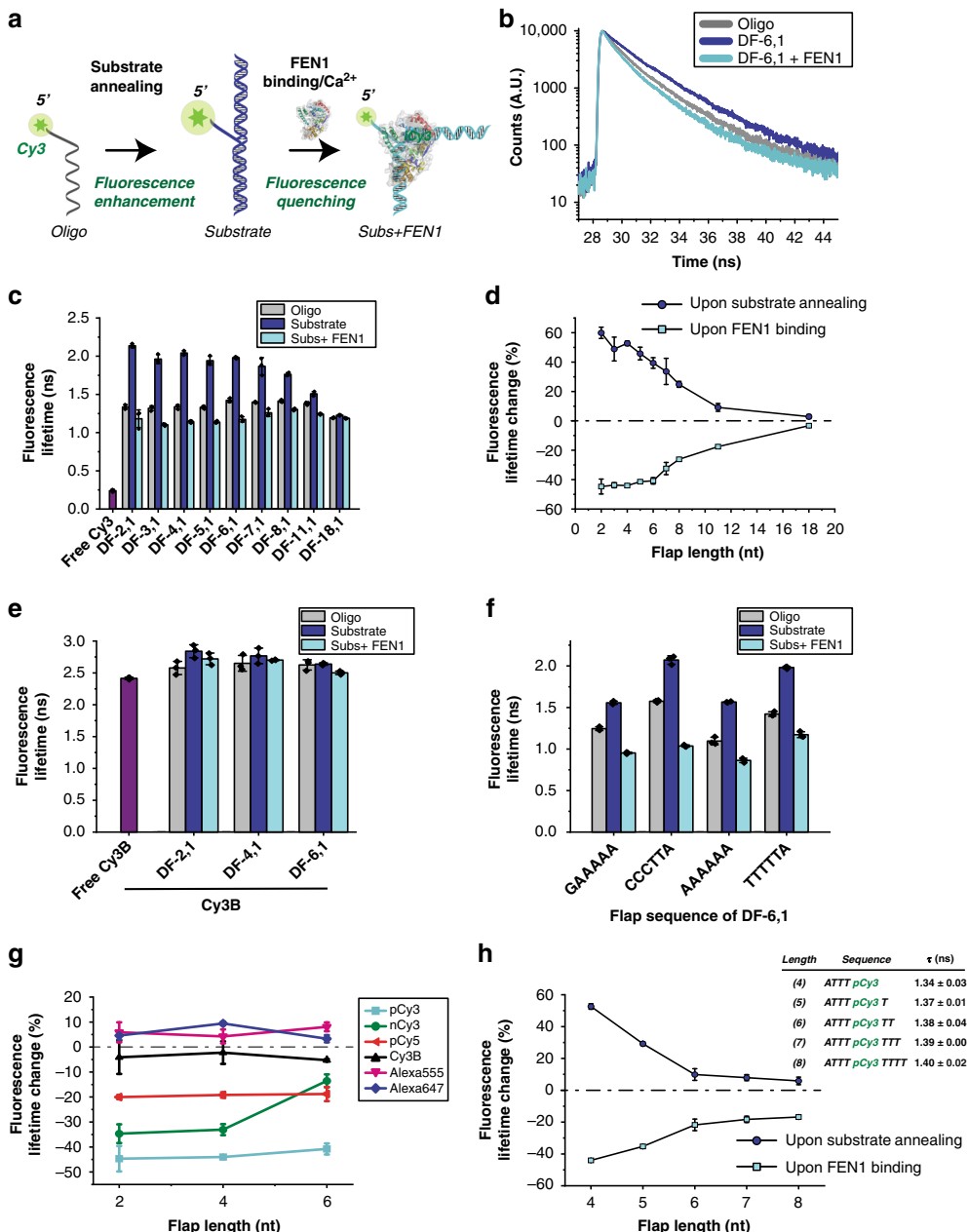

**Fig. 2** DNA-mediated bidirectional fluorescence modulation in the FEN1/DF system. **a** Schematic showing the oligo bearing pCy3 at its 5′-tip, annealed to two other oligos to form a double flap structure (DF) that serves as a substrate for FEN1. The ssDNA flap part of the DF is threaded through a narrow pathway in FEN1 for cleavage to happen. **b** Time-resolved fluorescence lifetime decays of Cy3 in oligo alone (gray), upon making the DF substrate (blue) and upon addition of FEN1 to the DF substrate (cyan) with a 6 nt-long 5′-flap. **c** Bar chart showing time-resolved fluorescence lifetime of Cy3 in oligo alone, upon making the DF substrate and upon addition of FEN1 to the DF substrate for flap lengths 2–18 (color scheme is same as **b**). Fluorescence lifetime of free Cy3 dye is included (in purple) for comparison. **d** Graph showing percentage of change in fluorescence lifetime, upon annealing of the DF substrate and FEN1 binding to the annealed substrate as a function of flap length. **e** Bar chart graph showing the time-resolved fluorescence lifetime of Cy3B-labeled oligos alone, upon making the DF substrate and upon addition of FEN1 to the DF substrate for flap lengths 2, 4, and 6. Cy3B is linked to the DNA via a 13-carbon linker. **f** Bar chart showing the time-resolved fluorescence lifetime of Cy3 in the oligo alone, upon making the DF substrate, and upon addition of FEN1 to the DF substrate for 6 nt-long 5′-flap with different flap sequences. **g** Graph showing the percentage of change in fluorescence, based on lifetime measurements in DF substrates for different fluorophores and different linkers, with and without FEN1. pCy3 is linked to DNA via 3-carbon linker. Other fluorophores used are: Cy3B, nCy3, pCy5, Alexa Fluor 555, and Alexa Fluor 647. **h** Graph showing the percentage of Cy3 fluorescence change upon annealing of the substrate and addition of FEN1, when pCy3 is incorporated inside the ssDNA flap. The inset table lists the flap length, sequence, and fluorescence lifetimes. Error bars and ± represent standard deviation (SD) from three replicates. Oligo sequences are listed in Supplementary Table 1

these interactions, as reflected by the local rotational diffusivity of the dye, and the photoisomerization rate from the excited trans state (Supplementary Fig. 2h-inset), suggesting that the dye's freedom and its accessibility to photoisomerization are strongly related.

To probe the effect of immediate nucleobases on NAIFE and PIFQ, we fixed the length of the 5′-flap but varied its sequence, and showed that although the lifetime of pCy3-DF was sequence-dependent, FEN1 quenched pCy3 fluorescence in all constructs (Fig. 2f). This supports our findings that PIFQ, as defined from

the protein binding perspective, is essentially mediated by the DNA and that FEN1 simply reverses NAIFE.

Following that, we characterized the effect of fluorophore types and their attachement chemistries on PIFQ. Using DF substrates of variable 5′-flap lengths, we examined FEN1-induced fluorescence modulation in the case of Alexa Fluor 555, Alexa Fluor 647 and Cy3 (nCy3) coupled via NHS, as well as Cy5 coupled through phosphoramidite (pCy5) at the tip of the 5′-flap. We showed that pCy3, nCy3, and pCy5 exhibited a significant PIFQ, while Alexa Fluor 555 and Alexa Fluor 647 only exhibited a moderate to low PIFE, respectively (Fig. 2g). This result underscores the fact that the observed quenching effect is not a pCy3-specific phenomenon. The difference in fluorescence modulation among these fluorophores can be attributed to differences in the length of their polymethine bonds, their overall charge and the presence/absence of sulfonate groups (Supplementary Fig. 1).

We next investigated the potential impacts of changing the position of the fluorophore in the 5′-flap. Coupling pCy3 to the tip of the 5′-flap restricted its rotational freedom as shown by its higher lifetime relative to the free Cy3 dye (Fig. 2c). We then restricted pCy3 further from both sides by placing it internally at position 4 in the 5′-flap while gradually adding 1 nt to its 5′-side. We observed that both NAIFE and PIFQ decreased gradually with the addition of one or two dTs and plateaued at a lower level with the extra additions (Fig. 2h). We therefore postulate that pCy3 when restricted only from one side or slightly from the other side (1 dT) might be capable of forming interactions with the neighboring nucleotides, which would be lost with further restrictions, leading to an overall reduction in fluorescence modulations. These interactions, if present, would reduce the rate of photoisomerization from the excited trans conformer.

Collectively, these experiments suggest that the interactions of pCy3 with the neighboring DNA is the main cause of NAIFE upon substrate annealing and FEN1 binding breaks these interactions leading to the observed PIFQ. Moreover, they suggest that the photophysical properties of Cy3 in a particular substrate are affected by the overall DNA-Dye structure and the interactions within.

**Initial lifetime of Cy3-DNA sets the stage for PIFE or PIFQ.** To explore the versatility of PIFQ and its adaptability to other systems, we investigated the interaction of human ssDNA-binding protein RPA with ssDNA[34,35] (Fig. 3a). This system removes the peculiarity of DF NAIFE and offers a general DNA–protein binding system. We designed a library of pCy3-labeled oligos with varying sequences, fluorophore positions, and fluorophore types (Supplementary Table 1).

Our library included 16 3′-pCy3-labeled oligos with varying sequences and initial fluorescence lifetimes that ranged between 0.5 and 1.5 ns (Fig. 3b). Upon RPA binding, all oligos exhibited PIFE to a varying degree. However, their final fluorescence lifetimes in the presence of RPA seemed to average out around a threshold of 1.66 ns, regardless of their initial lifetimes. In general, the degree of PIFE seemed to depend on the initial fluorescence lifetime of the oligo itself.

On the 5′-side, the initial fluorescence lifetimes of 21 pCy3-labeled oligos ranged from 0.7 to 2.2 ns (Fig. 3c). This library exhibited versatile RPA-induced fluorescence modulations involving fluorescence enhancement, quenching or insignificant effect. Overall, we observed that the oligos initial fluorescence lifetime seemed to dictate RPA-induced fluorescence modulation with a significant anti-correlation between the initial lifetime and fluorescence change. In most cases, oligos with lifetimes of 1 ns or below showed PIFE, whereas oligos with lifetimes of 1.3 ns or higher showed PIFQ; oligos with lifetimes in-between did not

show any significant change. Nevertheless, irrespective of the type of effect, RPA appeared to bring the oligos fluorescence lifetime to a defined threshold of 1.27 ns.

The library of 15 internally pCy3-labeled oligos displayed diverse initial lifetimes that ranged from 0.9 to 2.6 ns (Fig. 3d). Surprisingly, O-328, with an initial lifetime of 2.6 ns (Fig. 3d), displayed an even longer fluorescence lifetime than that of Cy3B (2.4 ns; Fig. 2e). This library showed versatile fluorescence modulations, albeit with majority of PIFE. Similarly to the 3′- and 5′-libraries, the RPA-induced fluorescence modulation seemed to bring the oligos fluorescence lifetime to a higher defined threshold (1.78 ns). Furthermore, steady-state fluorescence measurements confirmed RPA-induced fluorescence modulations of oligos in the three libraries (Supplementary Fig. 3c).

Next, we added one more dimension to the diversity of our library by varying fluorophore types and positions. We tested seven different fluorophores each placed at six different positions. Except for Cy3B, all fluorophores contained a polymethine bond capable of undergoing cis-trans photoisomerization (Supplementary Fig. 1). It is worth noting that the sequence of these oligos is, in principle, preserved; however, changing the position of the fluorophore entails changing the local sequence sensed by the fluorophore. RPA induced PIFE in most fluorophores at different positions with the exceptions of Cy3B and 5′-end-pCy5. This PIFE was significant for pCy3, nCy3, and Alexa Fluor 647 (20–80%) but highly elevated for Alexa Fluor 555 (80–160%) (Fig. 3e). The dominant PIFE outcome might be a result of the experimental design where only one particular sequence was used, probably one with low-initial fluorescence lifetime.

Having established that the initial fluorescence lifetime of the DNA–Dye complex sets the stage for RPA-induced fluorescence modulation, we sought to investigate whether other ssDNA-binding proteins induce similar pCy3 fluorescence modulations (Fig. 3f–h). We measured fluorescence changes upon binding of *E. coli* SSB or T7 gp2.5 to each of the oligos in the three libraries and compared them to that of RPA. These measurements confirmed that the average final lifetimes upon protein binding were protein-specific. Nevertheless, the general trend that the type of fluorescence modulation is governed by the difference between the variable DNA-Dye's initial state and the protein-specific final state applies. The difference in the average final lifetimes between the different proteins, however, affected the amplitude of the change and in some cases even the direction of the change; these findings can be further visualized using box chart plots (Supplementary Fig. 3a).

Finally, similar to the case of FEN1/DF system, we performed time-resolved fluorescence anisotropy measurements on a subset of terminally labeled oligos. The results indicate a strong correlation between the dye's local rotational freedom and its photoisomerization rate in the initial state (Supplementary Fig. 3b).

**The structural properties of the DNA–Dye complexes.** This set of experiments is designed to further investigate the initial fluorescence state of a library of oligos, by probing the role of different external modulators on the overall DNA-Dye structure and rotational freedom in the absence of protein. Assuming that PIFE is an analogous effect to the presence of a viscogen, we investigated the effects of varying the glycerol concentration on the fluorescence lifetime for several DNA constructs (Fig. 4a). Although most of these constructs, except Cy3B, responded to an increase in glycerol concentration by increasing their lifetime, they exhibited a wide range of initial lifetimes (lifetimes at 0% glycerol) and a varied response to an increase in viscosity. However, the final lifetimes (lifetimes at 100% glycerol) for all

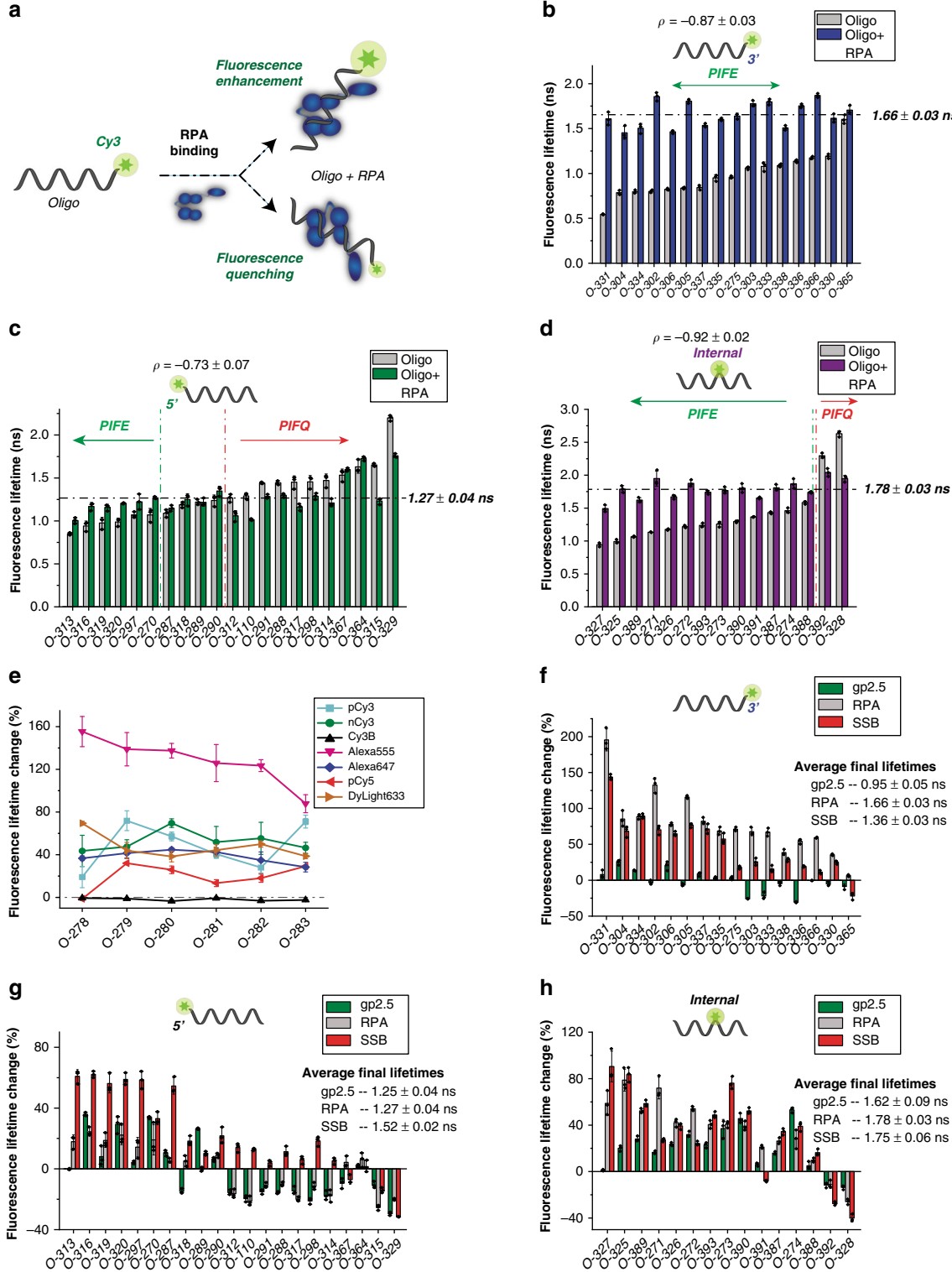

constructs seemed to approach a similar value, as photo-isomerization would be fully inhibited at a high viscogen concentration.

For a more quantitative approach, the dynamic viscosity[36,37], converted from glycerol percentage (0–60% range; Supplementary Fig. 4a), was plotted against the fluorescence lifetimes of the various DNA constructs (Supplementary Fig. 4b). The resulting curves were fitted with a Michaelis–Menten-like hyperbola, where $K_{1/2}$ represents the dynamic viscosity at which half of the maximum fluorescence lifetime is achieved. Generally, constructs

with a broader dynamic range of lifetimes resulted in higher $K_{1/2}$ values, resulting from the significant difference between their initial lifetimes and maximum lifetimes (Supplementary Fig. 4b). When the initial lifetimes are plotted against the inverse of the $K_{1/2}$ values, a striking linear dependence is observed (Fig. 4b). This linear dependence clearly highlights the three dye coupling classes, i.e., free dye or DNA-coupled from either one or two sides. The initial lifetime is generated by a rate that can be further decoupled into two components: photoisomerization-dependent and photoisomerization-independent rates (see SI Methods). The

**Fig. 3** The initial lifetime of Cy3-DNA sets up the stage for PIFE or PIFQ. **a** Schematic describing the effect of RPA binding to a pCy3-labeled oligo with different sequences and different fluorophore positions, leading to either quenching or to a fluorescence enhancement. **b** Fluorescence lifetimes of the 16 3′-pCy3-labeled oligo library with different sequences, either with RPA (blue) or without (gray). The dashed line shows the average fluorescence lifetime in the presence of RPA. The Pearson coefficient of the correlation between the initial fluorescence lifetime (in the absence of RPA) and the change in fluorescence (%) is reported with its standard error. **c** Library of 22 5′-pCy3-labeled oligos with their fluorescence lifetimes in the absence (gray) and presence (green) of RPA is shown (as described in **b**). The green vertical dashed line delimits oligos that show an overall PIFE effect, upon addition of RPA, whereas the red vertical dashed line delimits those showing an overall quenching effect. **d** Bar chart showing the fluorescence lifetimes of a library consisting of 15 internally pCy3-labeled oligos in the absence (gray) and the presence of RPA (purple). Horizontal dashed line, Pearson coefficient, green and red vertical dashed lines are as described in **c. e** Representation of the effect of the fluorophore type and position (six different positions with O-278 as 5′-labeled, O-279– > O-282 internally labeled and O-283 3′-labeled). The changes in fluorescence (%), upon the addition of RPA, are shown for the six oligos with different fluorophore types and linking chemistries (pCy3 in cyan, nCy3 in green, Cy3B in black, Alexa Fluor 555 in magenta, Alexa Fluor 647 in blue, pCy5 in red and DyLigth633 in orange). The horizontal dashed line represents the zero line. Fluorescence lifetime change (%) upon addition of gp2.5 (green), RPA (gray) or SSB (red) to oligos of 3′-pCy3-labeled library (**f**), 5′-pCy3-labeled library (**g**), and internal-pCy3-labeled library (**h**). The average final lifetimes upon protein additions are reported. Error bars represent SD from three replicates and ± represent standard error of the mean (SEM). Oligo sequences are listed in Supplementary Table 1

photoisomerization-independent rate exhibited a ~1.3-fold difference between free Cy3 and the class of pCy3 constructs, which accounted for a slight increase of pCy3-coupled DNA lifetime (Supplementary Fig. 4c). On the other hand, the photoisomerization-dependent rate ($k_{iso}$) from the excited trans state, in pure water, showed a ~20-fold difference between free Cy3 and the class of pCy3 constructs (Supplementary Fig. 4d). Taken together, these two observations showed that the variance in the initial lifetime between different constructs stemmed mainly from the photoisomerization pathway.

We next investigated the effects of the overall structure of an oligo–dye complex, given a particular sequence, linker, and fluorophore position, on the fluorescence lifetime of the fluorophore. We hypothesized that annealing a complementary strand to an oligo would disrupt the structure formed by the oligo–dye complex, just as a protein-binding would. We tested this hypothesis with a subset of internally pCy3-labeled oligo library (Fig. 3d). The annealing of their complementary strands induced fluorescence modulations (NAIFE for enhancement or NAIFQ for quenching; Fig. 4c) that followed the general principle observed with ssDNA/ssDNA-binding protein system. This demonstrates that fluorescence modulations need not necessarily associate with a specific protein binding, but rather to the molecule that propagates the interaction, in this case the pCy3-labeled oligos. Moreover, the anti-correlation between the initial lifetime and the fluorescence change still dictated the direction of the fluorescence modulation.

In the case of environmentally sensitive fluorophores, the notion of a single lifetime is replaced by a broad landscape of lifetimes accessed by this fluorophore in different environments and configurations. This was previously characterized using a microarray of short DNA oligos labeled with Cy3 or Cy5 assayed for their relative intensities[18,19,38]. However, these studies were limited, as the only variable characterized was the short DNA sequence. Here, we compiled a landscape of Cy3 accessible lifetime values, from a very significant number ($N = 398$) of lifetime measurements, encompassing diverse contexts such as DNA sequence, linker chemistry, buffer conditions, DNA conformation, and protein association (Fig. 4d). This landscape stretched between two extreme values, that of free Cy3 in water and that of O-328 (18) in RPA buffer.

Finally, since O-328 stood out as having the highest Cy3 fluorescence, we further probed the basis of its high fluorescence. We postulated that such high fluorescence could be due to the rigidification of the excited trans state imposed by the overall DNA structural configuration and possibly the additional effect of other pathways beyond photoisomerizaton. To test the rigidification hypothesis, we systematically perturbed this structural

configuration by trimming down the size of O-328 two nucleotides at a time, from 22 nts (O-328) down to 8 nts (Fig. 4e). The lifetimes of these oligos remained largely consistent, up to a length of 16 nts, beyond which the lifetimes dropped down gradually. Upon annealing the corresponding complementary strands, the formed dsDNA exhibited a constant lower lifetime across different lengths. Taken together, these results indicate that a specific DNA structure may form within the core of O-328 (central 16 nts) within the context of RPA buffer. We postulated that the buffer viscosity, ionic strength, pH and/or divalent metals could affect the DNA–Dye structure, and consequently the dye's fluorescence. We measured the fluorescence lifetime of O-328 in solutions containing each of the buffer components separately (Fig. 4f). Surprisingly, O-328 lost its extraordinarily high fluorescence in water and other solutions except for the one containing 50 mM KCl, suggesting that potassium ions induce the formation of a secondary structure within O-328.

We further investigated the photophysical properties of O-328 to understand better the peculiarity of its high fluorescence beyond inhibition of photoisomerization. O-328 absorbance and emission spectra showed no significant spectral shifts in all Cy3 contexts (free, random oligo, O-328, with or without K$^+$) (Supplementary Fig. 4e). Decoupling the de-excitation rates (Supplementary Fig. 4f) suggests that the super-enhanced fluorescence in O-328 upon K$^+$ binding as compared to Cy3B, most likely stems from a slightly prolonged radiative decay and a reduced rate of non-radiative pathways beyond photoisomerization.

**PIFQ can be used at the single-molecule level**. To explore the applicability of PIFQ to single-molecule techniques, we sought to characterize FEN1's catalytic efficiency, using a single-labeled DF. In our previous work, we used DF-6,1 with pCy3 placed at the tip of the flap and Alexa Fluor 647 downstream of the nick junction[25,27,28]. With this labeling scheme, FEN1 binding and DNA bending causes FRET shift from 0.8 to 0.48 for few frames, before FEN1 cleaves the pCy3-labeled 5′-flap, leading to the instantaneous loss of the 5′-flap and consequently FRET (Fig. 5a and Supplementary Fig. 5a)[25,27]. The distribution of the dwell times spent in the bent state, fitted to a gamma distribution, averaged at ~163 ms, and was used to estimate the single turnover catalytic rates[25].

As shown in Fig. 2, FEN1 binding to DF-6,1 induced a 40% quenching of the pCy3-labeled 5′-flap fluorescence. This PIFQ could be a useful single-molecule cleavage assay, since it can reduce the complexity of the smFRET cleavage assay, especially when choosing an optimized FRET pair, detectable FRET change, and acceptors' stability. Furthermore, the cleavage by FRET

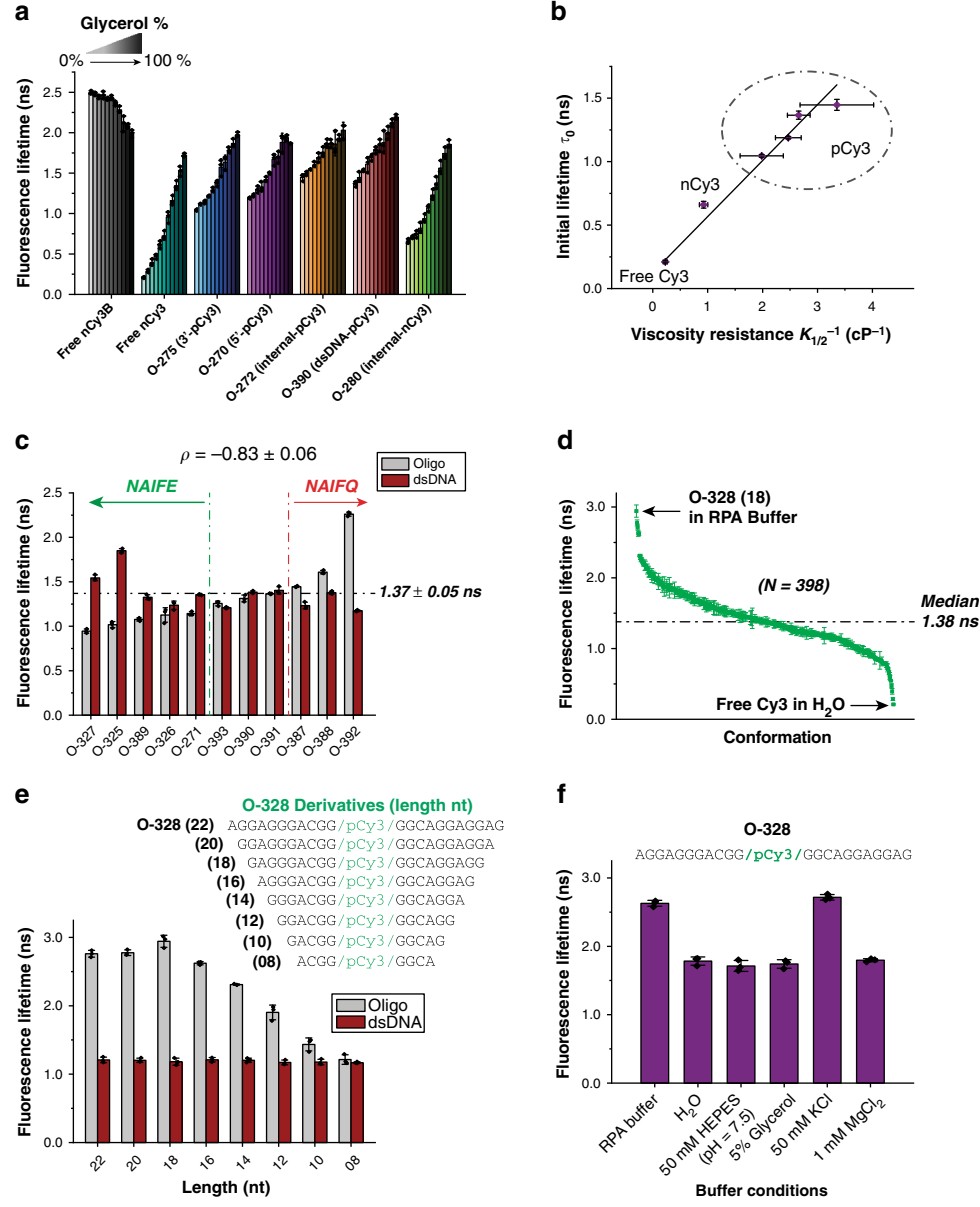

**Fig. 4** Insights into the structural properties of the DNA–Dye complexes. **a** Effect of viscosity on fluorescence; bar charts represent the fluorescence lifetimes of different fluorophores (free or DNA–Dye complexes), at increasing concentrations of glycerol (0–100%), in 10% increments. **b** Dependence of the fluorescence lifetime, without glycerol, on the viscosity resistance. The plot shows a linear dependence with slope of 0.44 ± 0.02 ns cP and y-intercept of 0.13 ± 0.04 ns. The horizontal error bars represent the standard error of the mean. The goodness of the linear fit, $R^2$ value, is 0.989. **c** Fluorescence lifetimes of 11 internal-pCy3-labeled oligo library with different sequences (gray) and their corresponding dsDNA (red). The horizontal dashed line shows the average fluorescence lifetime of dsDNA in this library. The green vertical dashed line delimits oligos that show an overall NAIFE effect, upon annealing of their complementary strand, whereas the red vertical dashed line delimits those showing an overall NAIFQ effect. The Pearson coefficient of the correlation between the initial fluorescence lifetime (ssDNA) and the fluorescence change (%), upon annealing the complementary oligo, is reported with its respective standard error. **d** Fluorescence lifetime landscape of Cy3 compiling all the Cy3 lifetimes measured in this study ($N = 398$). The horizontal dashed line represents the median value of the landscape. **e** Bar chart indicating the fluorescence lifetime of O-328 and its derivatives, in ssDNA (gray) and dsDNA (black) forms, measured in RPA buffer. **f** Bar chart indicating the fluorescence lifetime of O-328 in ssDNA form, in different individual RPA buffer components. Error bars represent SD from three replicates and ± represent SEM. Oligo sequences are listed in Supplementary Table 1

reports on all post DNA bending catalytic steps, whereas cleavage by PIFQ may report on specific catalytic steps only. In smPIFQ assay (Fig. 5b), FEN1 binding causes a single-step quenching before the signal is lost (Fig. 5b and Supplementary Fig. 5b). This quenching step is interpreted as FEN1 recognizing its substrate, while the loss of pCy3 signal marks the instantaneous 5′-flap release. Our confidence in attributing the pCy3 loss to a cleavage of the 5′-flap, rather than to photobleaching, is reinforced by the fact that pCy3 is stable in the absence of FEN1 under similar

buffer conditions[25]. Moreover, in both smFRET and smPIFQ, the changes in FRET and fluorescence, respectively, are significantly distinguishable from noise as illustrated in Fig. 5d and described in SI Methods. Hence, even a change occurring in one frame can be confidently assigned to a FRET or PIFQ change.

Surprisingly, the distribution of dwell times before cleavage from smPIFQ showed a single-exponential decay (Fig. 5b), suggesting a single rate-limiting step, in contrast to the rise and decay distribution observed using smFRET cleavage assay

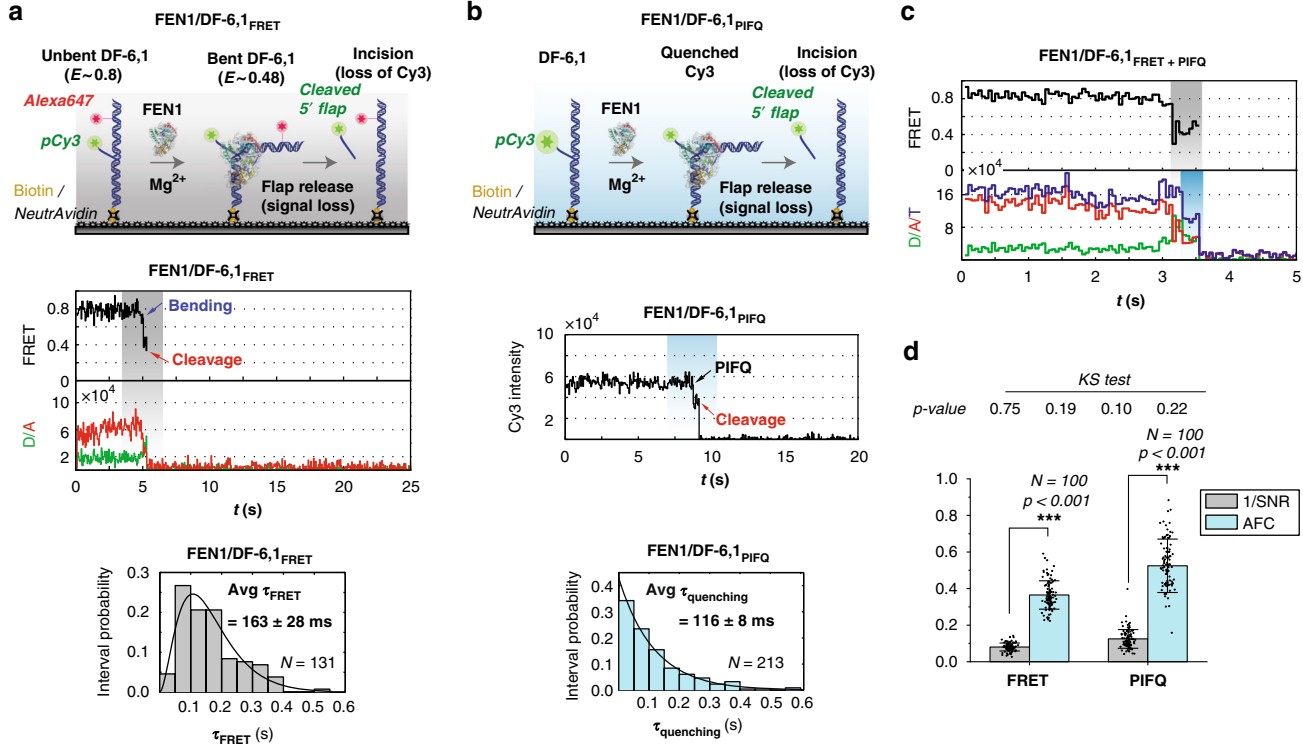

**Fig. 5** PIFQ can be used at the single-molecule level to infer FEN1 catalytic kinetics. **a** smFRET cleavage assay used to follow the catalytic kinetics of FEN1 on DF-6,1 substrate. Top: schematic representation of the experimental approach. Bottom: representative single-molecule trace of a cleavage event showing the FRET change and the following loss of signal. The distribution of the time spent in bent state for $N = 131$ cleavage events was plotted and fitted with gamma distribution. The mean and standard error of the mean are reported. **b** Single-molecule cleavage assay (smPIFQ) with a single label (pCy3) placed at the tip of the flap. Top: schematic showing the design of the assay. Bottom: representative single-molecule trace monitoring the fluorescence quenching of pCy3 upon interaction with FEN1 and subsequent signal loss. The time spent in the quenched-state is quantified for $N = 213$ events and its distribution is plotted and fitted to an exponential decay, with the mean and standard error of the mean reported. **c** Representative smFRET trace showing the FRET change (highlighted in gray) and the total intensity change (highlighted in cyan) before the signal loss. The total intensity change corresponds to the quenching of the Cy3 signal. **d** Bar chart showing the reciprocal of the signal-to-noise ratio (1/SNR-shown in gray) and the absolute fractional change (AFC-shown in Cyan) (light blue) for smFRET and smPIFQ cleavage traces as described in SI Methods section. The error bars correspond to the standard errors of the mean (SEM) of the 1/SNR and AFC in each case. The analysis was performed using $N = 100$ traces in both cases. The *p*-value ($p < 0.001$) is calculated for estimating the statistical difference between 1/SNR and AFC using a two-sample *t*-test for both smFRET and smPIFQ cleavage traces. The inset table shows the *p*-values generated by the Kolmogorov–Smirnov (KS) test of normality for the four data sets. The KS test failed to reject the null hypothesis that the data is normally distributed as indicated by $p > 0.05$. Error bars represent SD of $N = 100$ traces and ± represent SEM. Measurements for **a**–**c** were recorded with 50 ms temporal resolution

(Fig. 5a). Furthermore, the average dwell time from smPIFQ (116 ms) was slightly shorter than that from smFRET (163 ms). We propose that the difference in the average dwell times and the distribution patterns of these dwell times, obtained via the two assays, can be interpreted such that smPIFQ and smFRET start reporting at different time points of the FEN1 cleavage reaction. It may be possible that FEN1 induces quenching of Cy3's fluorescence, only once the flap has been fully threaded and positioned, whereas smFRET starts reporting on the DNA bending, prior to the threading step. These results reveal an important mechanistic information regarding the timing of the 5′-flap threading, in relation with DNA bending[25]. More importantly, this shows that smPIFQ can be used to question different aspects of this cleavage pathway, hence providing an extra dimension and adding to the information already acquired from smFRET assays[25,27].

Moreover, FRET and PIFQ can be monitored simultaneously in the same smFRET cleavage experiment, as was reported earlier for smFRET/smPIFE[39], by following the change in FRET as well as the change in total fluorescence intensity of both donor and acceptor. This allowed us to distinguish between the starting points of both phenomena. The time traces showed the lower

FRET state spanning over the average dwell time of 163 ms (Fig. 5a). However, during this dwell time, two distinct steps are distinguishable: one where total fluorescence intensity was preserved and the second where total fluorescence intensity dropped (Fig. 5c and Supplementary Fig. 5c). The second step corresponds to the donor quenching just before cleavage. The occurrence of FRET change before PIFQ supports our interpretation that PIFQ most likely reports on a step involving 5′-flap threading following DNA bending.

Our next system focused on K+-induced fluorescence enhancement of O-328 (Fig. 4f). We hypothesized that K+ altered the overall structure of the Cy3-labeled oligo, perhaps through the formation of some secondary structure, similar to G-quadruplexes case[40]. It is then expected that the addition of RPA, a protein known to melt secondary structures[41], would quench the highly fluorescent O-328/K+ and attempt to restore the original fluorescent state of O-328. However, this hypothesis posits one caveat; perhaps, the addition of K+ induces the formation of a dimer, rather than a secondary structure in the monomer. To test this hypothesis, we resorted to a single-molecule assay that uses a longer oligo containing the sequence of O-328 at one end and annealed to a biotinylated oligo at the other

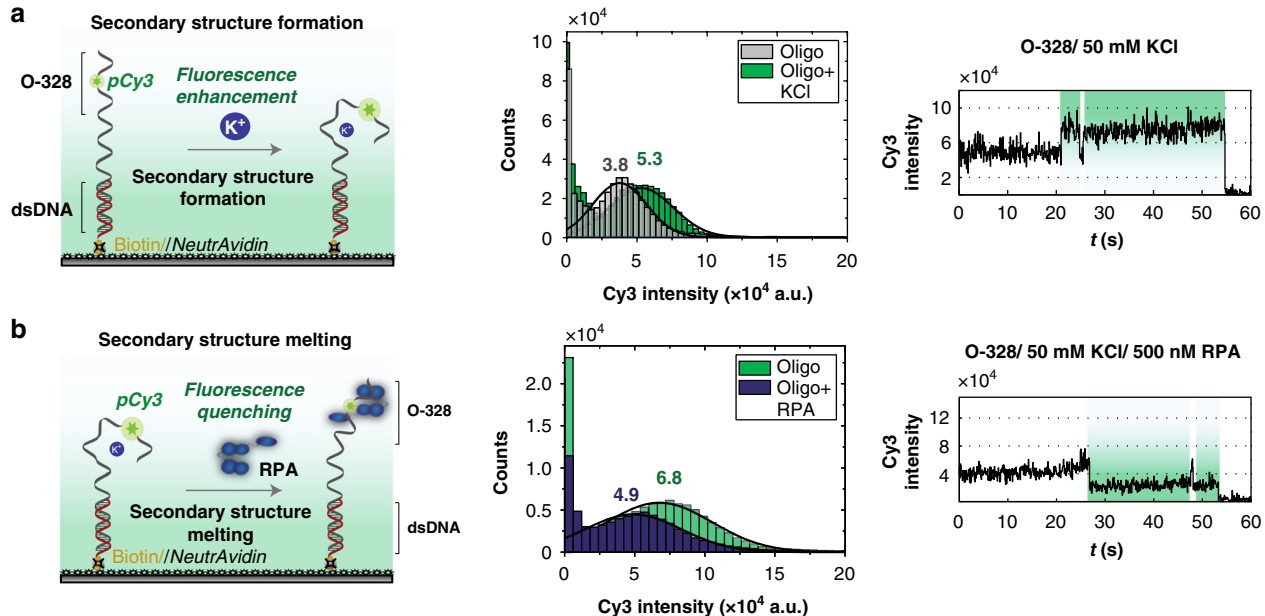

**Fig. 6** DNA structural changes monitored using Cy3 fluorescence modulation. **a** Secondary structure formation. Left: schematic showing the experimental set-up. Upon the addition of $K^+$, a secondary structure is formed leading to an enhancement of Cy3 fluorescence. Middle: Cy3 fluorescence intensity histograms with (green) $K^+$. The histograms are fitted to Gaussian distributions. Right: representative single-molecule trace showing the transitions to the enhanced-fluorescence state, upon formation of the secondary structure, as a result of the binding of $K^+$. **b** Melting of the secondary structure. Left: schematic drawing showing the DNA construct used in **c** with a secondary structure formed in the presence of $K^+$. However, upon addition of RPA, this secondary structure is melted, leading to the quenching of Cy3's fluorescence. Middle: Intensity histograms of the Cy3 fluorescence, in the absence (green) and presence (blue) of RPA. Histograms are fitted with Gaussian distributions. Right: representative single-molecule trace showing the transitions to the quenched-fluorescence state, upon melting of the secondary structure induced by the binding of RPA measurements for **a** and **b** were recorded with 100 ms temporal resolution

end, creating a primer/template (P/T) junction. This ensured that the fluorescent end of O-328 was free and away from the surface. Such design provided a well-spaced surface, where the formation of a dimer was almost impossible. In the absence of $K^+$, the P/T junction showed a low-fluorescence intensity profile, which was clearly shifted to a higher fluorescence regime upon the addition of 50 mM KCl (Fig. 6a), similarly to what we observed using bulk lifetime measurements (Fig. 4f). More importantly, the time traces upon the injection of KCl (Fig. 6a and Supplementary Fig. 5d) showed an initial instantaneous transition followed by a stable higher fluorescence state. This Cy3 intensity transition is interpreted as a dynamic transition between two states in a two-state binding system. This indicates that the $K^+$-induced fluorescence enhancement is a single-step process (within our temporal resolution) rather than a progressive process that goes through multiple states to reach its maximum. Moreover, the time traces exhibited one-step photobleaching events, further confirming the monomer nature of the P/T junction (Fig. 6a and Supplementary Fig. 5d).

Next, we tested whether RPA could melt this $K^+$-induced secondary structure. Using the same P/T junction, we confirmed the high-intensity profile of this substrate in the presence of KCl, which got quenched upon the addition of RPA (Fig. 6b). This indicates that RPA is capable of melting the secondary structure and reversing the $K^+$-induced fluorescence enhancement to a lower intensity profile. Time traces (Fig. 6b and Supplementary Fig. 5e) showed quenching in a single-step upon RPA binding, as well as single-step photobleaching events.

Taken together, these single-molecule assays show how fluorescence modulation, if controlled, can be used to understand diverse biochemical phenomena, at the single-molecule level. Both single-molecule FEN1 cleavage and the monitoring of $K^+$-induced structural changes in O-328, as well as its disruption by RPA, are examples that can be studied in a simple way, using a single fluorophore.

## Discussion

Understanding biological processes using fluorescence tools, whether at the single-molecule level, or at the ensemble-level, may benefit from simpler approaches that do not require many extraneous labels. Increasingly, fluorescence modulation upon binding of an interacting-partner or conformational changes within a single biomolecule is being leveraged to gain valuable knowledge of a multitude of biological phenomena and processes, and to elucidate the underlying mechanisms. In the context of protein binding to dye-labeled DNA, PIFE has been widely used in both single-molecule- and ensemble-assays. Furthermore, fluorescence modulation within a single biomolecule has also been used to understand their conformational changes[42].

The dependence of fluorescence parameters of Cy3 and similar dyes on the physical environment has proven to be advantageous in the design of many assays including PIFE. These parameters are mainly a product of Cy3 cis-trans photoisomerization in the excited state. Within these bounds, there are two possible restriction scenarios: (1) Cy3 fluorescence is the product of a rate competition between photoisomerization upon excitation and fluorescent de-excitation from the excited trans state, for a fraction of time when the fluorophore is free to rotate and thereby free to photoisomerize, and (2) for a fraction of time, Cy3 has permanent interaction with neighboring residues (DNA bases and amino acids), which would lead to a more rigidified excited state that is unable to rotate or photoisomerize. These scenarios need not be mutually exclusive and Cy3 fluorescence can be a product of both the rate competition and the totally rigidified excited state to a different extent. These two scenarios are likely to

be the cause of the multi-exponential nature of the fluorescence de-excitation of DNA-coupled cyanine dyes[10].

In this study, our FEN1/DF system showed that an opposite effect of PIFE exists. This PIFQ effect was preceded by a NAIFE upon substrate formation (Fig. 2c, d) with clear anti-correlation, suggesting that quenching itself is due to the modulation of the high Cy3-DF fluorescence. Experiments with Cy3B-labeled DF strongly suggest that both fluorescence modulations are mediated by photoisomerization (Fig. 2e). We further investigated the quenching process with multiple parameters, including Cy3 labeling or using different fluorophores.

Our results suggest that the initial fluorescence state is a key factor that determines whether PIFE or PIFQ occurs. This was clearly evident in a library of pCy3-labeled oligos with diverse initial fluorescence states, which experienced PIFE or PIFQ upon RPA (and other ssDNA-binding proteins) binding (Fig. 3b–h). Most importantly, the fluorescence change anti-correlated with the initial fluorescence state of Cy3-DNA (Fig. 3b–d). Taken altogether, the data suggest that binding of ssDNA-binding proteins to ssDNA does not actively modulate Cy3 fluorescence to a similar outcome (i.e., fluorescence increase, decrease or no change), but rather the outcome itself is dependent on the initial fluorescence. This initial fluorescence state of the DNA–Dye complex is dictated by its overall structure as imposed by various molecular interactions of the dye with the DNA.

These interactions can be related to steric hindrance imposed by the DNA structure on the dye, as well as specific interactions such as electrostatic, hydrophobic, π-π stacking and hydrogen bonding[14]; for example, in the context of 5′-labeled DNA, combination of these interactions may give rise to 5′-stacking[8,31–33]. Two theoretical models were developed to explain fluorescence enhancement based on the concept of steric hindrance and the available volume for dye rotation-photoisomerization[12,16]. These studies characterized Cy3 excited-state behavior in terms of ratios of available volumes for rotation based on the proposed relation between rotation and photoisomerization[12–15]. Generally speaking, these models can be extrapolated to explain our observed modulations. However, for these models to hold in the case of quenching, they should allow for the dye in the initial state to be more restricted than in the final state. For instance, an oligo with a considerable secondary structure or comparatively strong specific interactions that restrict the dye's photoisomerization in the initial state can be disrupted in the presence of ssDNA-binding protein or a complementary strand, resulting in a relative increase in the dye photoisomerization.

On a separate note, our results drew our attention to the use of PIFE (or PIFQ) as a protein-binding assay, in particular when associating PIFE (or PIFQ) with a positive protein binding at a specific site. Here we advise special caution to be exercised when interpreting such fluorescence modulations. For instance, ssDNA oligos with lengths supporting RPA binding but with sequences exhibiting an initial fluorescence around that of the average final Cy3-labeled DNA–RPA complex, did not show any significant change upon the addition of excess RPA (Fig. 3b–d). Hence, without a proper preliminary assessment, one could falsely attribute the lack of fluorescence change to a poor (or even non-existent) affinity of RPA to these DNA sequences. In such scenarios, one could possibly change the sequence to generate a fluorescence modulation, or to amplify the existing one.

To our knowledge, a systematic comparison of the commonly used fluorophores in order to determine their relative predisposition and degree of protein-induced fluorescence change has not been reported. In this study, we conducted such comparison with FEN1/DF (Fig. 2g) and RPA/ssDNA (Fig. 3e) systems to establish such parameters for commonly used fluorophores. Except for Cy3B, all other fluorophores displayed a modulation of

the fluorescence, upon protein binding. Interestingly, in ssDNA/ RPA system, Alexa Fluor 555 showed an extreme PIFE effect of >100%, suggesting that it might be an ideal environmentally sensitive fluorophore, and hence quite powerful for PIFE studies (Fig. 3e).

The preceding experiments highlighted the role of the initial state of a system in fluorescence modulation, which probed further investigation of this initial state in protein-free Cy3-DNA system. This investigation showed that when Cy3 is more rigidly bound to the DNA, its lifetime displays less sensitivity to the changes in the environmental conditions, thus exhibiting a less dynamic change in its fluorescence (Fig. 4a). It also showed that when Cy3 becomes more rigidly bound to the DNA, its initial lifetime increases. Indeed, the relationship between $1/K_{1/2}$ and the initial lifetime, under various Cy3 conditions, showed a linear dependence (Fig. 4b). The slope of this linear relationship can then be used as a quantitative parameter to evaluate any fluorophore's propensity to change its fluorescence, with the change of environmental factors. Furthermore, we showed that dye's local rotational freedom strongly correlates with its photoisomerization rate (Supplementary Fig. 3b).

It is worth mentioning that a previous study used transient absorption spectroscopy to monitor photoisomerization of cyanine dyes and its role in PIFE phenomenon[4]. By directly monitoring post-excitation cis-isomer formation, the authors concluded that the effect of protein presence in PIFE leads to a reduction in the photoisomerization pathway. We believe that these results can be extrapolated to our observed fluorescence modulations, including quenching upon protein binding or DNA annealing. Nevertheless, we point out that a decrease in the fluorescence lifetime need not be associated only with the formation of cis-isomer since isomerization to the twisted $S_1^*$ is enough to compete with the fluorescent de-excitation from trans excited state, without the requirement of branching into the cis-isomer. To gain insight into the degree of photoisomerization in our systems, we have performed increasing viscosity experiments to gradually inhibit photoisomerization, including trans $S_1^*$ to the twisted $S_1^*$.

The fluorescence lifetimes of Cy3 generated under all the different conditions yielded a comprehensive landscape that provides a valuable insight into Cy3 fluorescence lifetime bounds (Fig. 4d). This landscape showed ~15-fold difference in lifetime between the upper and lower bounds and a median of 1.38 ns. Altogether, the viscosity dependence (Fig. 4b) and the fluorescence landscape (Fig. 4d) demonstrate the ease and range of Cy3 fluorescence modulation, upon changing the environment, with several practical implications. For instance, one would choose a condition that yields a fluorescence state at the bottom of the lifetime landscape, if the preferred outcome is to generate a PIFE effect, and vice versa for PIFQ. Likewise, conditions generating fluorescence lifetimes around the median of the landscape could be favored, if no photophysical effect upon protein binding is required, for example, in the case where the fluorophore will be used for accurate FRET measurements to monitor conformational changes[43]. Sequences of desired initial lifetime could be used from the library presented in the current work or from other high-throughput sources[18,19,38]. We propose, in future experiments, to identify patterns in labeled DNA sequences and to correlate them with their corresponding structures and consequently their lifetimes.

Experiments with ssDNA/ssDNA-binding proteins (Fig. 3f–h), as well as ssDNA/dsDNA (Fig. 4d) systems emphasized that the initial fluorescence of pCy3-oligos most likely stemmed from their overall structures, which have the potential to be disrupted by an external modulator (ssDNA-binding proteins or complementary oligo). Nonetheless, the varied average final lifetimes in either case highlights the role of the external modulator in the

final outcome (enhancement, quenching or no change). The super-enhanced fluorescence of O-328 was particularly intriguing and probed a series of investigations (Fig. 4e, f and Supplementary Fig. 4e, f) to understand its underlying photophysics. We expect that this sequence can be integrated into other sequences to generate PIFQ.

Using multiple single-molecule experiments, we showed how our understanding of the fluorescence modulations could be applied to rationally design PIFE or PIFQ experiments to study catalytic kinetics, conformational changes and protein binding at the single-molecule level. As a proof of concept, our smPIFQ FEN1 cleavage assay with a singly labeled DF monitored FEN1 cleavage reaction, in real time, and allowed us to access FEN1 catalytic kinetics (Fig. 5) in a similar fashion to smFRET assay[25,27]. However, smPIFQ cleavage assay tracked the reaction at different steps from those observed with smFRET, making PIFQ and smFRET complimentary assays, rather than redundant. The combined smFRET/smPIFQ assay corroborated the findings of the individual assays. On the other hand, single-molecule assays using a singly labeled P/T junction, within the context of O-328 sequence, allowed us to monitor the formation of the $K^+$-induced secondary structure, and its melting by RPA, in real time (Fig. 6). Thus, we demonstrated how PIFQ, similar to PIFE, could be used to study conformational changes as well as protein binding.

With a combination of time-resolved fluorescence lifetime, steady-state fluorescence measurements and single-molecule assays, we have shown that Cy3, and by reasonable extrapolation all the other environmentally sensitive dyes, has the unique property of context-dependent fluorescence identity. This property could be significantly leveraged by careful considerations of the fluorescence parameters in the initial state of the system being studied. If the initial fluorescence state of the Cy3-conjugated system can be manipulated by changing the parameters affecting Cy3 fluorescence to occupy either extremity of Cy3 fluorescence landscape, fluorescence enhancement or quenching can be achieved.

## Methods

**DNA oligos and substrates**. DNA oligos were custom-synthesized with their respective modifications by Integrated DNA technologies (IDT) or Sigma-Aldrich. Oligos with phosphoramidite coupled Cy3 (pCy3) or Cy5 (pCy5) were directly purchased from IDT or Sigma. However, oligos used for Cy3B, Alexa Fluor 647, Alexa Fluor 555, nCy3 and DyLight633 labeling were ordered from IDT harboring site-specific amine-modified thymine. All DNA oligos were HPLC purified and the list of these oligos is shown in Supplementary Table 1. We would like to note that the notation of each oligo number does not bear any sequence or specific property significance, but it is rather for our lab inventory purposes. The monofunctional NHS dyes were purchased from GE healthcare. For the labeling reactions, the dyes were dissolved in DMSO to a final concentration of 20 mM while the modified oligos were dissolved in DNAse free $H_2O$ to a final concentration of 0.2 mM. The reactions proceeded by mixing the oligos at a final concentration of 0.05 mM with ~ 40-fold molar excess of the different dyes in freshly prepared labeling buffer containing 50 mM $Na_2[B_4O_5(OH)_4]$ (pH 8.5) followed by 6 h incubation at room temperature in the dark with gentle mixing. Ethanol precipitation was then used to precipitate the labeled oligos out of the excess dyes. Further purification of the labeled oligos was performed using denaturing polyacrylamide gel electrophoresis (PAGE). The labeling efficiency was calculated by comparing the absorbance of DNA at 260 nm and that of the respective dye at its absorption maximum wavelength. Oligos were labeled with > 90% labeling efficiency.

Double flap (DF) substrates were constructed by annealing the three strands, template: 5′-flap: 3′-flap in a 2:1:4 molar ratios, in TE-100 buffer (50 mM Tris–HCl pH 8.0, 1 mM ethylenediaminetetraacetic acid (EDTA) pH 8.0, 100 mM NaCl). For annealing, a thermocycler PCR machine was used where the oligos mixture was heated at 95 °C for 5 min followed by step cooling at a rate of 1 °C per 1 min down to 25 °C. Similarly, dsDNA and P/T junction substrates were constructed by mixing labeled oligos with a ratio of 1:3 in TE-100 buffer and annealed using the same thermocycling method. Substrates were purified over non-denaturing PAGE and their purity was assessed to be > 95%.

### Protein expression and purification. The gene sequences encoding human FEN1, *E. coli* SSB and T7 bacteriophage gp2.5 were cloned via Gibson assembly into a

modified pE-SUMO-pro expression vector (Lifesensors) harboring an N-terminal double $His_6$-Tag followed by SUMO fusion protein. These clones were transformed into *E. coli* BL21 (DE3) strain (NEB), and expressed in 2xYT media by 0.2 mM IPTG induction. Cells were harvested and then lysed in buffer A (50 mM Tris–HCl pH 7.5, 5% (v/v) glycerol, 750 mM NaCl, 10 mM β-mercaptoethanol (BME) and 30 mM imidazole). Purification was carried out by two sequential Ni-NTA columns, separated by SUMO protease cleavage. Proteins were eluted with a linear gradient against buffer A + 300 mM imidazole, where in the second Ni-NTA chromatography the proteins went into flow through. Proteins were then concentrated and further purified over HiLoad superdex-75 pg (for human FEN1 and T7 gp2.5) or HiLoad superdex-200 pg (for *E. coli* SSB) size exclusion columns. Size exclusion chromatography was performed using a buffer containing 50 mM HEPES-KOH pH 7.5, 500 mM NaCl, 2 mM dithiothreitol (DTT), and 10% (v/v) glycerol. Fractions containing the desired proteins were collected, flash frozen, and stored at −80 °C.

For human RPA, the plasmid encoding all three RPA subunits (pET11d-tRPA) was a generous gift of Prof. Marc S. Wold[44]. The protein was expressed in BL21 (DE3) cells in TB media at 37 ˚C after 0.3 mM IPTG induction. Cells were collected and resuspended in lysis buffer (HI Buffer + 1 mM phenylmethane sulfonyl fluoride (PMSF)); HI buffer contained 30 mM HEPES-KOH pH = 7.8, 0.25% (w/v) myo-inositol, 0.25 mM EDTA-pH = 8.0, 1 mM DTT and 0.01% (v/v) igepal (NP-40). RPA was purified over HiTrap Blue HP 5 mL column (GE healthcare) with extensive washing using increasing salt concentrations and finally eluted with a linear gradient against HI Buffer + 1.5 M NaSCN. Next, fractions containing the 3-subunit complex were desalted and concentrated over a Hydroxyapatite (Bio-Rad) column against an elution buffer of HI Buffer + 50 mM potassium phosphate (pH = 7.8). The final step of purification was performed over MonoQ column (GE healthcare) and the protein was eluted with linear gradient against HI Buffer + 400 mM KCl. Fractions containing the 3-subunit RPA complex were collected, concentrated, flash frozen in liquid nitrogen and stored at −80 °C.

Protocatechuate 3,4-dioxygenase (PCD) encoding subunits *pcaH* and *pcaG* were codon optimized and cloned with individual promoters in pRSF-1b vector. An N-terminal decahistidine tag was added to pcaH subunit of the PCD enzyme to aid the purification. PCD expression clone was transformed into *E. coli* BL21 (DE3) strain, and expressed in 2xYT media by 0.2 mM Isopropyl β-D-1-thiogalactopyranoside (IPTG) induction at 25 °C for 8 h. Ferrous ammonium sulfate was added just prior to induction at the final concentration of 20 mg/L of culture. PCD was purified using Ni-NTA affinity and Superdex-200 pg gel filtration chromatography.

**Time-resolved fluorescence lifetime measurements**. Time-resolved fluorescence lifetime measurements were carried out using QuantaMaster 800 spectrofluorometer (Photon Technology International Inc.) equipped with a Fianuim supercontinuum fiber laser source (Fianium, Southampton, U.K.) operating at 20 MHz repetition rate. Arrival time of each photon was measured with a Becker-Hickl SPC-130 time-correlated single photon counting module (Becker-Hickl GmbH, Berlin, Germany). Measurements were collected under magic angle (54.7°) conditions and photons were counted using time to amplitude converter (TAC). To reduce the collection of scattered light, a longpass filter (550 nm) was placed at the emission side. In all measurements, 10,000 counts were acquired. The instrument response function (IRF) was estimated using a Ludox colloidal silica suspension dissolved in water.

Measurements were recorded at room temperature in FEN1 reaction buffer for DF substrates-related samples, in RPA buffer for RPA-related samples, or other buffers as indicated in the corresponding sections. Samples containing green fluorophores (pCy3, nCy3, Alexa Fluor 555, and Cy3B) were excited at 532 nm and emission was collected at 565 nm with 5 nm slit width for both excitation and emission. On the other hand, samples containing red fluorophores (Cy5, Alexa Fluor 647 and DyLight633) were excited at 632 nm and emission was detected at 665 nm with similar slit widths (5 nm) for both excitation and emission. The fluorophores lifetime decays were then obtained using FluoFit software package (PicoQuant) applying the IRF and fitted to one or two-exponential decays. The one-exponential decay fit was used for free fluorophores while the two-exponential decay fit was utilized for all other samples. The best fit was chosen based on reduced chi-square and randomness of the residuals.

The reported lifetimes are the mean of amplitude-averaged lifetimes of three independent replicates. The fluorescence change based on the lifetime measurements was calculated as the percentage of the difference between the final lifetimes upon protein addition to DF substrate (in case of FEN1) or oligo (in case of RPA) and the initial lifetimes of these DNA constructs as compared to the initial lifetimes, keeping in mind the sign of the change. The Pearson correlation coefficient reports the correlation between this percentage of fluorescence lifetime change and the initial lifetime. This coefficient of the correlation was calculated using bootstrap statistics[45].

Experiments probing the effect of viscosity on the fluorescence lifetimes of fluorophores were performed according to the methods described above. Each studied fluorophore, free or attached to DNA, was dissolved in increasing concentrations of glycerol (0–100% v/v in increments of 10%) diluted in DNAse free water. These increasing concentrations of glycerol were used to determine the dynamic viscosity of each sample[36,37], keeping in mind the measurements were

done at standard temperature and pressure. Fluorescence lifetimes of the different samples were then plotted against the calculated dynamic viscosity and the curves were fit with a Michaelis–Menten type hyperbola as described in SI Methods. The inverse of $K_{1/2}$ (a measure of a fluorophore's rigidity in a particular context) was plotted against the fluorescence intensity in the absence of glycerol and the data points were fitted to a linear dependency.

**Single-molecule fluorescence measurements**. Single-molecule measurements were all performed at room temperature in a custom airtight microfluidic flow cell with a glass coverslip that was functionalized and passivated by 1:100 molar ratio of biotinylated polyethylene glycol (Biotin-PEG-SVA MW 5000) and polyethylene glycol (mPEG-SVA MW 5000) (Laysan Bio Inc.). DNA substrates (100–200 pM) were immobilized onto the surface using biotin–NeutrAvidin interaction. Prior to the DNA immobilization, the surface was incubated with 0.2 mg/mL NeutrAvidin for 10–15 min followed by excessive washing with reaction buffer to remove excess NeutrAvidin and block any extra unspecific binding sites. To enhance the fluorophores photostability and reduce photo-blinking, our imaging buffer included a mixture of reaction buffer, a protocatechuic acid (PCA)/protocatechuate-3,4-dioxygenase (PCD) oxygen scavenging solution[46] and 2 mM Trolox (Sigma-Aldrich). All single-molecule experiments were performed using a custom-built TIRF-FRET set-up[47]. Several movies of each condition were recorded on different fields of view in two-color alternating excitation (2c-ALEX)[48] mode and/or continuous excitation mode. The time resolution for the different experiments is mentioned in their respective figure legends. Data was extracted using twotone software[49] and analyzed using Matlab and OriginPro.

FEN1 nuclease assays of DF substrates were performed using double-labeled substrates for smFRET measurements or using single-labeled DF substrates to observe smPIFQ. In both experiments, 250 nM FEN1 was used to cleave the substrates in reaction buffer containing 50 mM HEPES-KOH pH 7.5, 100 mM KCl, 5% (v/v) glycerol, 10 mM MgCl$_2$, 0.1 mg/mL bovine serum BSA, and 1 mM DTT. Recording using continuous excitation of green laser was initiated prior to the arrival of FEN1 to the flow cell. The movies were recorded under continuous flow of protein at a time resolution of 50 ms.

For smFRET experiments, a cleavage event was identified by the preceding decrease of FRET signal from E~0.8 to E~0.48 signifying the bending step, an essential step in FEN1-substrate recognition, followed by the loss of signal after the incision of the 5′-flap and the loss of the donor. This decrease in FRET signal is clearly distinguishable and the donor and acceptor signals anti-correlate. In this case, the time spent in the low-FRET state (bent state) ($\tau_{FRET}$) was calculated by manually counting the frames. The distribution of $\tau_{FRET}$ was plotted and fit to a gamma distribution using Matlab dfittool and the mean and standard error of the mean are reported.

Similarly, for smPIFQ experiment, a cleavage event was identified by the preceding quenching of pCy3 signal signifying FEN1 engaging the flap substrate and followed by the loss of pCy3 signal after the incision of the flap. This quenched-state step is clearly distinguishable as it only occurs when followed by the complete loss of signal. Likewise, the cleavage event and loss of signal is distinguishable from pCy3 photobleaching as it is always preceded by a quenched-state step. The time spent in the quenched-state ($\tau_{quenching}$) was calculated by manually counting the number of frames. The distribution of $\tau_{quenching}$ was plotted and fit to a single-exponential decay using Matlab dfittool and the mean and standard error of the mean are reported.

For this set of experiments, the DNA substrate used is a P/T junction composed of a long (82 nt) pCy3-labeled oligo containing O-328 (22) sequence at its 5′-end and annealed to a biotinylated complimentary short (22 nt) oligo at its 3′-end. This substrate is immobilized to the surface through biotin–NeutrAvidin interaction with dsDNA region near the surface and the ssDNA region containing O-328 (22) extending further away from the surface. For monitoring the secondary structure formation, first, the DNA substrate (100–200 pM) was immobilized to the surface in RPA reaction buffer excluding KCl. Three movies of different fields of view were recorded at equilibrium using continuous excitation of green laser. Second, RPA reaction buffer containing 50 mM KCl was injected into the flow cell. Prior to the arrival of the KCl-containing buffer to the flow cell, recording was started under continuous flow of buffer. Finally, three movies of different fields of view were recorded at equilibrium with the exchanged buffer was reached. The movies taken at equilibrium before and after the injection of 50 mM KCl were used to construct the distributions of pCy3 intensity in the two conditions. These distributions were fit with Gaussian peaks using OriginPro and the center of these peaks are reported. The movie recorded under flow was used to monitor the change of Cy3 fluorescence, in real time, as shown in the time traces.

Similarly, to observe the melting of the secondary structure, the same P/T substrate was immobilized to the surface in RPA reaction buffer containing 50 mM KCl. Three movies were recorded, at equilibrium, before the injection of 100 nM RPA in the RPA reaction buffer. A movie was recorded, starting prior to RPA arrival to the flow cell and under continuous flow. At last, three movies were recorded after the final equilibrium with RPA was reached. These movies were used to construct the pCy3 intensity histograms and time traces, in a similar fashion to those described for the formation of the secondary structure.

**Reporting summary**. Further information on research design is available in the Nature Research Reporting Summary linked to this article.

## Data availability
Data supporting the findings of this manuscript are available from the corresponding author upon reasonable request. A reporting summary for this Article is available as a Supplementary Information file. The source data underlying Figs. 2c–h; 3b–h; 4a–f and 5a, b, d and Supplementary Figs 2b–h; 3a–c and 4b–d,f are provided as a Source Data file.

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

## Acknowledgements

This work was supported by King Abdullah University of Science and Technology under Grant URF/1/3432-01-01 to S.M.H. We are grateful to Prof. Stefan T. Arold (KAUST) for providing access to time-resolved fluorescence spectrophotometer. We thank Daniela-Violeta Raducanu for her support in some experiments. We thank Yujing Ouyang for preparation of functionalized coverslips and Afnan Shirbini for some of the RPA used in this study. We also thank members of Samir M. Hamdan's lab for helpful discussions.

## Author contributions

F.R. and S.M.H. conceived the project. F.R., V.S.R., M.S.Z. and S.M.H. designed, planned, and analyzed the experiments. F.R., V.S.R., M.S.Z. and M.T. performed the experiments. F.R., V.S.R., M.S.Z., S.H. and S.M.H. discussed the results and wrote the manuscript.

## Additional information

**Competing interests:** The authors declare no competing interests.

