## [Peer Review File · Nature Communications]

Reviewers' comments:

Reviewer #1 (Remarks to the Author):

In general, I like to suggest the authors to change the nomenclature. Based on the data, protein-binding does not really quench the fluorescence of fluorophores; it just sometimes enhances it less than nucleic acids do. So maybe the authors can consider a name like nucleic acid-induced fluorescence enhancement (NIFE). And the difference in enhancement by nucleic acids and proteins can be used as a signal readout in single-molecule measurements. The following are some other minor points I would like the authors to address.

1) The authors said "We observed that Cy3N behaved similarly to iCy3, but that the magnitude of fluorescence enhancement, upon formation of the DF substrate and its subsequent quenching by FEN1, was significantly reduced (Fig. 2g)." Fig. 2g is only about the magnitude of fluorescence enhancement upon FEN1 binding, not upon substrate annealing.

2) The authors tested 16 3'-iCy3-labeled oligos with varying sequences in Fig. 3b. Each of those sequences has a notation, e.g. "O-331". It is helpful to what the notations represent in the supplementary information.

3) When the authors mentioned "the degree of fluorescence enhancement seemed to depend (correlate) on the initial fluorescence lifetime of the ssDNA oligo", it would be nice to show some data on the degree of fluorescence enhancement, such as those histograms shown in Fig. 5c,d.

4) The authors said "We propose that the difference ..., can be interpreted by imagining that smPIFQ and smFRET start reporting at different time points of the FEN1 cleavage reaction." It is not scientific to use word like "imagine", and this part of the manuscript is just generally very hand-waving.

5) The time resolution is 50 ms in Fig. 5a,b, and the bin size used in the histograms is also about 50 ms. It means the events in the first bin is just one data point in the movie. I am not sure if this one frame intensity drop can be easily distinguished from fluctuation.

6) Can the authors comment on the transient intensity drops and jumps after adding potassium and RPA, in the trajectories shown in Fig. 5c,d?

7) The authors can remove the fitting parameters (i.e. the slope and intercept) in Fig. 4c and mention them in the text and legend. (edited)

Reviewer #2 (Remarks to the Author):

The manuscript “Initial state of the DNA-dye complex sets the stage for on demand fluorescence modulation upon protein binding” by Rashid, Raducanu & Zaher et al., is a very interesting thorough examination of the current rationalization behind protein-induced fluorescence enhancement (PIFE), or should I say more generally Cy3-induced fluorescence change. This work coins a new approach, named protein-induced fluorescence quenching (PIFQ) that occurs oppositely to the expected enhancement of Cy3 fluorescence by the vicinity of a bound protein. This manuscript presents many important new insights that question the existing explanation of PIFE. Overall this is an important experimental validation, and might I say quite a thorough one. However, although I appreciate the work performed to produce this manuscript, I do have some general concerns.

The manuscript reports on a variety of experiments that involve molecular binding, some in the ensemble level and some at the single-molecule level, some with internally labeled Cy3 and some with it labeling either one of the DNA termini. The collection of all findings point to the fact that Cy3 fluorescence changes are not only dictated by protein binding, but also by their interactions with the DNA when protein is not bound. This is not at all surprising. PIFE referring to the effect of protein binding to change Cy3 fluorescence is now well-established as only one reason for fluorescence change out of others. This was actually established in recent papers, some of which referred to it as nucleic-acid induced fluorescence enhancement (NAIFE), however were not mentioned (“An atomistic view on carbocyanine photophysics in the realm of RNA”, by Börner and co-workers).

The authors are right in that also the interactions of Cy3 with DNA in the protein-free state, are important. However, eventually, PIFE (or any kind of Cy3 fluorescence enhancement) involves comparing Cy3 fluorescence, each time between two states. In PIFE it is between when Cy3 is labeling DNA and when a protein binds this DNA. In NAIFE it is comparing a ssDNA (or RNA) with Cy3 before and after it hybridizes to a complementary strand. So of course the fluorescence of Cy3 is compared in these two cases, hence these Cy3 fluorescence change measurements are relative. Additionally, it is directly the ratio in the interactions of Cy3 with its surroundings in each of these two states that makes the difference. Largely speaking, the difference may be in the degree of steric restriction of Cy3 in each state, but it can also be the time Cy3 spends in a restrictive interaction in each state. The former explanation of changes in the steric restriction of the dye to slow down its photoisomerization is the mainstream explanation for Cy3 fluorescence changes. However, the authors choose to test a DNA system in which Cy3 is labeling the 5' terminus. It has already been proven that Cy3 labeling the 5' end of DNA stacks on top of it (Liu & Lilley, *Biophys. J.*, 2017; Oullet et al., *Biophys. J.*, 2011). By that, photoisomerization is halted until Cy3-DNA stacking interaction ceases temporarily. This, in turn, leads to more de-excitations from the trans isomer that was photo-

selectively excited (due to high dipole strength, compared to the negligible dipole strength of ground-state Cy3 in cis isomer), which eventually yields higher fluorescence. In fact, when moving from an oligo to a state in which it is hybridized to the substrate DNA, fluorescence increases, which can mean that now, not only stacking interactions sterically restrict the isomerization of the excited Cy3 but also the vicinity of the hybrid DNA. When FEN1 is added to the mix, fluorescence decrease. One thing for sure, stacking interactions of Cy3 to the terminus of the DNA to which it is bound are now inhibited by the protein surface. So overall, the protein vicinity here can have a double effect: 1) steric restriction of the freely-rotating-freely-isomerizing Cy3, which increases fluorescence; and 2) disruption of Cy3 stacking interactions with the 5' terminus of the DNA, which decreases fluorescence – leading to a net decrease in fluorescence. With this possible explanation the addition of bases to the Cy3 labeled base, so that it is now internally labeling DNA, and the decrease in fluorescence change is expected (Fig. 2h). The authors write “However, when the fluorophore is significantly restricted from both sides, we believe that some of these interactions may be lost, leading to a reduced effect of both the enhancement upon substrate formation and PIFQ”. However, this is merely the effect of removing the Cy3 stacking interaction to the terminus of the DNA. My suggestions for the FEN1 experiments are:

1. Mention the 5'-Cy3 stacking interactions as another source of complication that introduces a component of fluorescence enhancement.
2. To prove or disprove the alternative explanation given above it would be important to test all 3 states (oligo, bound to substrate, and with FEN1)
 - a. for whether photoisomerization occurs at all – by transient absorption experiments. If stacking interactions occur between Cy3 and the DNA 5'-terminus, then they should be long-lived, which will not lead to formation of cis isomer, for the Cy3 excited molecules.
 - b. for whether Cy3 is totally rotationally obstructed – by fluorescence anisotropy decays. If stacking interactions occur between Cy3 and the DNA 5'-terminus, then the fluorescence anisotropy should be decaying very slow, with no fast component, because the fluorophore lacks rotational freedom

I would like to note that in previous works (see Lerner & Ploetz et al., J. Phys. Chem. B, 2016), it was not the absolute theoretical value of the dye available volume that was compared to the PIFE results, but rather the ratio of theoretical dye available volumes in free dsDNA and in the presence of the bound protein. By that, these authors did take into account at least the expected steric restriction of Cy3 in BOTH states. It is important to note that these studies included solely Cy3 internally-labeling dsDNA, to minimize the addition of the dye-DNA stacking interactions. If the authors would like to be even more accurate, they can use the method developed by Börner and co-workers in the paper about NAIFE. In fact, if also specific dye-DNA interactions are involved, a better model can be developed, in which for a fraction of time, f_1 , the dye is stuck in trans isomer (an approximation) and in the remaining $1-f_1$ it is again acting as an excited-state molecular rotor where photoisomerization is inhibited by steric restriction. Although comparing dye available volumes of each case has been utilized for the rationalization of PIFE, it has not been put to the test in this work. This part is definitely missing in this work. Therefore, I request a comparison of the experimental results to the theoretical calculations of the isomerization/rotational restriction, using either one of the theoretical calculations proposed in the literature.

It is not true that reduction on fluorescence was not reported in the context of PIFE. It has been used to measure the extrusion of nontemplate bases in initially transcribing complexes in bacterial transcription initiation (Ploetz & Lerner et al. Sci. Rep., 2016). It is just that the fluorescence has decreased in this stage relative to RPo stage, and this was expected if the Cy3-labeled base got extruded out of the protein surface. I request the authors to emphasize the relativeness of the method.

Finally, I would like to mention that most importantly, Cy3 fluorescence changes, assessed from ensemble-averaged experiments, are not as accurate as should be. Such results only qualitatively teach the readers about PIFE or PIFQ, since it is hard to make sure all components were bound. Even a report of the fluorescence lifetime components of ensemble-averaged experiments, and how these components change between different states is insufficient, since Cy3 already has a multiexponential decay. In my view, the best test of PIFE versus PIFQ is with the single-molecule experiments, presented in the last part of the manuscript. Additionally, in light of the alternative explanation I gave of a possible dual effect of Cy3 labeling the 5'-terminus of DNA, the only outstanding PIFQ result is that with the O-328 construct.

Regarding the single-molecule experiments, I find it interesting that Cy3 fluorescence-change experiments are compared to independently acquired smFRET experiments. Why not perform both? PIFE and FRET can be combined, either using alternating laser excitation (Ploetz & Lerner et al., Sci. rep., 2016) or via immobilized assays looking both at FRET and at the sum of donor & acceptor fluorescence intensities (Craggs et al., Nucleic Acid Res., 2014). This way it can be shown unambiguously as a Cy3 intensity change and not FRET change.

Regarding the lifetime of Cy3 internally labeling O-328 that is higher than Cy3B and the claim that it is because of the rigidification formed by secondary structure formation. But still, this means that the Cy3 isomer that was stabilized is more fluorescent than Cy3B. Some other factor has influenced Cy3 to have different photophysical parameters than Cy3. Was there a spectral change associated with this finding? Can you elaborate on that?

Regarding the Cy3B control experiments, the authors write that

“To probe whether the fluorescence modulations in FEN1/DF system are analogous to those in PIFE, particularly with respect to the modulation of photoisomerization, we exchanged the iCy3 fluorophore with a Cy3B in our 3-context system of oligo, DF, and DF/FEN1 for flap lengths of 2, 4 and 6 nt. Cy3B is an analog of Cy3 dye, but with a rigid inter-heterocyclic construction rather than a rotationally flexible polymethine bond (Supplementary Fig. 1).”

and also

“In our study, the fluorescence lifetimes of the Cy3B-bearing oligos alone and their corresponding DF substrates either with or without FEN1 showed no difference (Fig. 2e).”.

Does Cy3B labeling the 5'-terminus, also stack to it as does Cy3? How much? Is it comparable to the stacking interaction of Cy3? I suggest to perform this comparison, using the same construct with Cy3B, only comparing on the basis of fluorescence anisotropy.

Minor:

1. “maybe be attributed to the different lengths” should be changed to “maybe attributed to the different lengths”
2. The authors wrote “Even though both isomers can absorb light upon excitation, only the trans S1* -to-trans S0 transition results in fluorescence emission. Furthermore, several lines of evidence support that the ground state is primarily trans 4,6-8, with some studies even considering it to be all trans 9,10”. This statement is inaccurate. The correct statement is that although both cis and trans isomers exist in ground-state, it is mostly the trans isomer that is excited and not the cis one due to large dipole strength of the trans isomer and the very low dipole strength of the cis isomer (see works from Marcia Levitus proving this using transient absorption experiments).
3. Regarding the Jablonski diagram in Fig. 1: if the 90-degree intermediate is tightly coupled to the ground-state, and if the dipole strength of the cis transitions is negligible, then why at all show the transitions into, out of and within the cis excited-state? It is rarely visited.

Reviewer #3 (Remarks to the Author):

In this manuscript the authors contributing to the understanding of fluorescent enhancement and quenching of labeled DNA in the presence of proteins. In particular, the authors argue that the phenomenon of protein-induced fluorescence enhancement is probably mostly a special case of a more general modulation of fluorescence that can also include quenching, which the authors term protein-induced fluorescence quenching. It is clear that both DNA and proteins can affect the rate of cis-trans photoisomerization, and hence fluorescence intensity, of many commonly used cyanine dyes. It is well known that the nucleobase sequence has a strong influence on this rate. Since the fluorescence of these dyes increases in the presence of nucleobases, it makes sense that the addition of proteins to the mix can either increase the fluorescence either by enhancing the dye DNA interaction or by an independent effect that also contributes to a greater effective local viscosity. While it may be the case that many proteins can further enhance fluorescence, it seems clear that if the DNA sequence (or structure) is already strongly enhancing, further enhancements are unlikely and more likely to be quenching.

I think this is a strong manuscript that addresses an important topic. DNA-dye-protein systems are extremely common in many experimental contexts and the framework and results in this manuscript are likely to be very helpful for many researchers in designing effective experiments and in avoiding experimental artifacts due to the high environmental sensitivity of common cyanine dyes such as cy3 and cy5.

The manuscript is very well written and presents the data clearly. The results are supported by an extensive set of appropriate and comprehensive experiments. In my opinion, the manuscript is appropriate for publication in Nature Communications as is or with minor revisions.

One aspect that I found unnecessarily confusing is the non-standard usage of iCy3 and iCy5 for dyes conjugated to the DNA with phosphoramidites, and Cy3N for dyes conjugated via NHS. In particular, the naming scheme differs from the one used within the sequences in Supp. Table 1, which uses iCy3 for internal labeling. I would encourage the authors to adopt a more intuitive naming scheme such as: 5'-Cy3 instead of iCy3 (and equivalent for Cy3 or 3'-end labeling), iCy3 for internal labeling with amidite, and 5'/i/3'-NHS-Cy3/5 for the various forms of labeling via dye-NHS.

Another detail that seems important is a short summary of the actual sequence dependence that was observed in the fluorescence. There are quite a few tested oligos, so it might be possible to say something about the effect of the specific identity of the nucleobases immediately adjacent to the dyes.

Reviewers' comments:

Reviewer #1 (Remarks to the Author):

In general, I like to suggest the authors to change the nomenclature. Based on the data, protein-binding does not really quench the fluorescence of fluorophores; it just sometimes enhances it less than nucleic acids do. So maybe the authors can consider a name like nucleic acid-induced fluorescence enhancement (NIFE). And the difference in enhancement by nucleic acids and proteins can be used as a signal readout in single-molecule measurements.

We thank the reviewer for this suggestion. We agree with the reviewer that we should adopt the nomenclature of nucleic acid-induced fluorescence enhancement (NAIFE) as reported earlier in the literature (Reference 13) or of the opposite quenching effect (NAIFQ) as reported here for the fluorescence enhancement/quenching upon DNA-DNA interactions. These changes have been implemented strictly throughout the revised manuscript.

However, we still used the original term PIFQ to describe the quenching observed from the perspective of the protein binding to its preferred DNA substrate (the DF substrate in the case of FEN1 and the ssDNA oligo in the case of ssDNA-binding proteins). We believe that maintaining a consistent analogy with that described in the literature for PIFE is important.

The following are some other minor points I would like the authors to address.

1) The authors said "We observed that Cy3N behaved similarly to iCy3, but that the magnitude of fluorescence enhancement, upon formation of the DF substrate and its subsequent quenching by FEN1, was significantly reduced (Fig. 2g)." Fig. 2g is only about the magnitude of fluorescence enhancement upon FEN1 binding, not upon substrate annealing.

We apologize for causing confusion in the description of this figure and we thank the reviewer for pointing our oversight out. The revised manuscript reflects the correct description of Fig. 2g and features only the fluorescence quenching upon FEN1 binding.

2) The authors tested 16 3'-iCy3-labeled oligos with varying sequences in Fig. 3b. Each of those sequences has a notation, e.g. "O-331". It is helpful to what the notations represent in the supplementary information.

We present a list of all sequences in Supplementary Table 1. However, the notation of each oligo is just for our internal lab-inventory reference and does not denote any

specific property of the sequence. We have now made an explicit statement in the Methods section and in the figure legends to clarify this point.

3) When the authors mentioned "the degree of fluorescence enhancement seemed to depend (correlate) on the initial fluorescence lifetime of the ssDNA oligo", it would be nice to show some data on the degree of fluorescence enhancement, such as those histograms shown in Fig. 5c,d.

The quoted statement refers to the ensemble-based time-resolved measurements of different individual oligo sequences as shown in Fig. 3b-d. These figure panels show the fluorescence lifetimes of the oligos both in their initial fluorescence state (unbound) or final state (upon RPA binding). To address the reviewer's comment, we present the fluorescence change upon RPA binding, along with those changes that occurred upon binding of other ssDNA binding proteins (gp2.5 and SSB, new data), as percentage change (Fig. 3f-h). Additionally, as a distribution, we plot the initial and final lifetimes of oligos from each library in box plots (Supplementary Fig. 3a).

4) The authors said "We propose that the difference ..., can be interpreted by imagining that smPIFQ and smFRET start reporting at different time points of the FEN1 cleavage reaction." It is not scientific to use word like "imagine", and this part of the manuscript is just generally very hand-waving.

We appreciate the reviewer's suggestion. We removed the word "*imagine*" in the revised manuscript. We would like to comment that this part of the manuscript is mostly intended to draw the general reader's attention to the possibility of using smPIFQ as a complementary assay to smFRET. Nevertheless, since both Reviewer 1 and 2 commented on this point, we looked further into the data for more mechanistic findings on FEN1's substrate specificity. We have now added Fig. 5c and Supplementary Fig. 5c, which show that smPIFQ and smFRET can be tracked simultaneously, while still providing complementary information and confirming our initial interpretation of the data.

5) The time resolution is 50 ms in Fig. 5a,b, and the bin size used in the histograms is also about 50 ms. It means the events in the first bin is just one data point in the movie. I am not sure if this one frame intensity drop can be easily distinguished from fluctuation.

We are confident that we can distinguish the FRET or Cy3 intensity change from the fluctuation, even within one frame, for the following reasons. In the smFRET cleavage assay, we followed the anti-correlated change in donor and acceptor to reflect a FRET change. On the other hand, the Cy3 intensity change in the smPIFQ cleavage assay was significantly distinguishable from noise, even if it lasted for only one frame. To illustrate our confidence in the first bin, we further analyzed the level of noise in the

smFRET and smPIFQ data for N=100 traces as well as the absolute fractional change (FRET or PIFQ). The results show that there is a marked statistical difference ($p < 0.001$) between the FRET (or PIFQ) change and the corresponding noise associated with those traces (Fig. 5d).

6) Can the authors comment on the transient intensity drops and jumps after adding potassium and RPA, in the trajectories shown in Fig. 5c,d?

We interpret these transient intensity drops and jumps (now Fig. 5e,f in the revised manuscript) as corresponding to dynamic transitions in a two-state system: bound and unbound states to K^+ or RPA as dictated by their binding kinetics. The revised manuscript includes a statement about these transient changes. It is not feasible to calculate the association and dissociation of these transitions since the bound state is long lived.

7) The authors can remove the fitting parameters (i.e. the slope and intercept) in Fig. 4c and mention them in the text and legend. (edited)

We have taken the reviewer's suggestion into consideration and edited the figure and text accordingly.

Reviewer #2 (Remarks to the Author):

The manuscript "Initial state of the DNA-dye complex sets the stage for on demand fluorescence modulation upon protein binding" by Rashid, Raducanu & Zaher et al., is a very interesting thorough examination of the current rationalization behind protein-induced fluorescence enhancement (PIFE), or should I say more generally Cy3-induced fluorescence change. This work coins a new approach, named protein-induced fluorescence quenching (PIFQ) that occurs oppositely to the expected enhancement of Cy3 fluorescence by the vicinity of a bound protein. This manuscript presents many important new insights that question the existing explanation of PIFE. Overall this is an important experimental validation, and might I say quite a thorough one. However, although I appreciate the work performed to produce this manuscript, I do have some general concerns.

We thank the reviewer for the positive feedback on the manuscript.

The manuscript reports on a variety of experiments that involve molecular binding, some in the ensemble level and some at the single-molecule level, some with internally labeled Cy3 and some with it labeling either one of the DNA termini. The collection of all findings point to the fact that Cy3 fluorescence changes are not only dictated by protein binding, but also by their interactions with the DNA when protein is not bound. This is not at all surprising. PIFE

referring to the effect of protein binding to change Cy3 fluorescence is now well-established as only one reason for fluorescence change out of others. This was actually established in recent papers, some of which referred to it as nucleic-acid induced fluorescence enhancement (NAIFE), however were not mentioned (“An atomistic view on carbocyanine photophysics in the realm of RNA”, by Börmer and co-workers).

The authors are right in that also the interactions of Cy3 with DNA in the protein-free state, are important. However, eventually, PIFE (or any kind of Cy3 fluorescence enhancement) involves comparing Cy3 fluorescence, each time between two states. In PIFE it is between when Cy3 is labeling DNA and when a protein binds this DNA. In NAIFE it is comparing a ssDNA (or RNA) with Cy3 before and after it hybridizes to a complementary strand.

So of course the fluorescence of Cy3 is compared in these two cases, hence these Cy3 fluorescence change measurements are relative. Additionally, it is directly the ratio in the interactions of Cy3 with its surroundings in each of these two states that makes the difference.

We thank the reviewer for this detailed comment. Up to this work, some earlier studies showed that DNA-Dye interactions may influence the initial state. The final state, on the other hand, is markedly characterized via PIFE. We cited these studies and we apologize for missing a citation to the paper by Börmer and co-workers (now added). However, to the best of our knowledge, no previous work provided experiments or suggested that the initial state with respect to the well-determined final state can dictate the outcome and/or the magnitude of the fluorescence modulation (PIFE, PIFQ, or no change) upon DNA-DNA or protein-DNA interactions. This is the unique finding of our work.

It is very important to note that given the arbitrary nature of the initial state and the protein-specific final state, fluorescence changes cannot be predicted, as suggested by the reviewer, but only experimentally determined. In other words, the correlation between the initial DNA state and the fluorescence modulation upon protein binding will not be obvious unless the protein-DNA final state is conserved and a systematic study with different initial states is performed. Using Cy3 at a specific position, we show that the magnitude of the fluorescence modulation is determined by changes in the initial state relative to the conserved final state. This finding takes fluorescence modulation from its current arbitrary characterization to a more systematic characterization in which fluorescence modulation (PIFE or PIFQ) at a given Cy3 position can be achieved on demand by measuring the average final state and varying the initial state primarily by changing the sequence around the Cy3 position.

Largely speaking, the difference may be in the degree of steric restriction of Cy3 in each state, but it can also be the time Cy3 spends in a restrictive interaction in each state. The former explanation of changes in the steric restriction of the dye to slow down its photoisomerization is the mainstream explanation for Cy3 fluorescence changes. However, the authors choose to test a DNA system in which Cy3 is labeling the 5' terminus. It has

already been proven that Cy3 labeling the 5' end of DNA stacks on top of it (Liu & Lilley, Biophys. J., 2017; Oullet et al., Biophys. J., 2011).

By that, photoisomerization is halted until Cy3-DNA stacking interaction ceases temporarily. This, in turn, leads to more de-excitations from the trans isomer that was photo-selectively excited (due to high dipole strength, compared to the negligible dipole strength of ground-state Cy3 in cis isomer), which eventually yields higher fluorescence. In fact, when moving from an oligo to a state in which it is hybridized to the substrate DNA, fluorescence increases, which can mean that now, not only stacking interactions sterically restrict the isomerization of the excited Cy3 but also the vicinity of the hybrid DNA. When FEN1 is added to the mix, fluorescence decrease. One thing for sure, stacking interactions of Cy3 to the terminus of the DNA to which it is bound are now inhibited by the protein surface. So overall, the protein vicinity here can have a double effect: 1) steric restriction of the freely-rotating-freely-isomerizing Cy3, which increases fluorescence; and 2) disruption of Cy3 stacking interactions with the 5' terminus of the DNA, which decreases fluorescence – leading to a net decrease in fluorescence. With this possible explanation the addition of bases to the Cy3 labeled base, so that it is now internally labeling DNA, and the decrease in fluorescence change is expected (Fig. 2h). The authors write “However, when the fluorophore is significantly restricted from both sides, we believe that some of these interactions may be lost, leading to a reduced effect of both the enhancement upon substrate formation and PIFQ”. However, this is merely the effect of removing the Cy3 stacking interaction to the terminus of the DNA.

5' stacking of cyanine against a terminal base of dsDNA (references mentioned by reviewer) or ssDNA (Sanborn et al., J. Phys. Chem B., 2007) has been shown using both structural and photophysical information. In our study, we can possibly have 5' stacking against the terminal base of ssDNA in both the ssDNA-binding protein and the FEN1/DF systems. In this case, we agree with the reviewer's double-effect model. However, we view the double effect as competing with each other with the outcome dictated by their relative strengths in the initial and final states. In support of this, we observed PIFE and PIFQ in the simplest experimental design using a library of 5'-Cy3-ssDNA oligos and ssDNA-binding proteins, which allows for varying both the initial and final states (please see revised Fig. 3, which also includes new data on two additional ssDNA-binding proteins, *E. coli* SSB and bacteriophage T7 gp2.5). In particular, although the oligos are the same, *E. coli* SSB primarily gives PIFE while RPA and gp2.5 mostly give PIFQ. The observation of PIFE in *E. coli* SSB suggests that the dominating effect of 5' stacking does not lead to a decrease, as proposed by the reviewer, while the observation of PIFE and PIFQ using the same oligos points to the competition between the double effects.

The reviewer built the 5' stacking argument by focusing on the FEN1-DF system. Therefore, we would like to highlight, using results in the original manuscript and newly added data, that even in this system, 5' stacking is not the dominant effect as follows:

- 1) We believe that certain interactions are formed in the hybrid DNA substrate that lead to a reduction in the de-excitation flux through the photoisomerization**

pathway. The reviewer pointed to the experiment with the addition of bases to the 5' end of Cy3 as supporting evidence for 5' stacking in ssDNA. However, we interpret the results of this experiment differently. The initial lifetimes of the ssDNA oligos, themselves with increasing numbers of capping bases, do not significantly vary. We apologize for not providing these numbers in the initial manuscript (they are added to Fig. 2h in the revised manuscript). Therefore, in this experiment, as more bases are added to the 5' end of Cy3, the fluorescence enhancement upon substrate formation is diminished because of a loss in the interactions between the dye and hybrid DNA and not due to 5' stacking removal.

- 2) To account for the possibility of 5' stacking in ssDNA, we compared a poly (dT) oligo labeling the 5' terminus (O-319) to a poly (dT) oligo labeling the 3' terminus (O-337). The lifetimes of these two oligos are comparable, 0.90 ns and 0.82 ns, respectively. We also mention that a poly (dT) oligo is less susceptible to 5' stacking (References 9 and 13). Hence, the effect of 5' stacking in poly (dT) sequences increases the lifetime by around 0.08 ns under our buffer conditions. On the other hand, the annealing of the DF substrate increases the lifetime by 0.7 ns, which is ~10 fold higher than the effect of 5' stacking.
- 3) In our 5' flap oligos, the standard sequence of the 5' flap part comprises poly (dT). Therefore, we compared the 5'-labeled poly (dT) oligo (O-319) to the 5' flap oligo. These two oligos are expected to have similar 5' stacking due to having the same terminal bases before Cy3. However, they exhibit different fluorescence lifetimes; O-319 has a lifetime of 0.90 ns while the 5' flap oligo has a lifetime of 1.35 ns. 5' stacking cannot explain this difference in lifetimes. The difference must come from the overall structure of the oligo as dictated by the sequence that explains this difference.
- 4) 5'-labeled long flaps (such as DF-18,1) have the same terminal bases as the short ones and therefore would be expected to have same 5' stacking. Nevertheless, in the substrate form, there is significant loss of fluorescence enhancement as the flap length increases.

In conclusion, we propose that in the case of the FEN1-DF system, interactions are formed but the dominant ones are those formed between Cy3 and the hybrid DNA, not the ones from stacking on top of the 5' base of the 5' flap. We also showed that the 5' stacking mechanism is not significant even in the case of the more general ssDNA-binding proteins system in which the same 5'-labeled oligo gave PIFE or PIFQ upon binding of different ssDNA-binding proteins (Figure 3g). Moreover, throughout this study, we do not focus on the molecular mechanism that modulates interactions of the fluorophore with the DNA but rather take the approach of including all these interactions and their effects as part of the overall structure of the DNA-dye complex to maintain the paper's appeal to the general audience of *Nature Communications*. We appreciate the reviewer's comments that stimulated us to improve our argument about the FEN1/DF system, to add new results on different ssDNA binding proteins, and to highlight this discussion in the revised manuscript including acknowledging that 5' stacking has a place in the overall dye-DNA 3D structure.

My suggestions for the FEN1 experiments are:

1. Mention the 5'-Cy3 stacking interactions as another source of complication that introduces a component of fluorescence enhancement.

We have taken the reviewer's point into consideration and revised the manuscript by mentioning 5' stacking interactions as among the contributing interactions that govern the initial fluorescence in oligo/substrate.

2. To prove or disprove the alternative explanation given above it would be important to test all 3 states (oligo, bound to substrate, and with FEN1)

- a. for whether photoisomerization occurs at all – by transient absorption experiments. If stacking interactions occur between Cy3 and the DNA 5'-terminus, then they should be long-lived, which will not lead to formation of cis isomer, for the Cy3 excited molecules.

The reviewer is correct in stating that performing transient absorption experiments might reveal the degree of photoisomerization. Similar work using transient absorption experiments has been done earlier (Reference 5) comparing Cy3-DNA and protein in the PIFE system. They showed that “*by monitoring the formation of the cis isomer directly, the enhancement of Cy3 fluorescence correlates with a decrease in the efficiency of photoisomerization, and occurs in conditions where the dye is sterically constrained by the protein*”. We believe that these results could be extrapolated to our data. Performing similar experiments is beyond the scope of the present study since our main purpose here is to get a measurable effect that can be used to answer biological questions and ease the interpretation of the PIFE results in particular. Nevertheless, we performed a glycerol titration study on oligo and substrate in the DF system and showed that considerable photoisomerization takes place in the oligo, while the interactions within the hybrid DNA substrate substantially reduce the photoisomerization (Supplementary Fig. 2d-inset). However, the same experiment is not feasible in the presence of FEN1 because the high viscosity might interfere with the protein's ability to bind DNA. We also approximated the saturating lifetime of the FEN1-DF complex to that of Cy3B and calculated the photoisomerization rate, which indicates that FEN1 binding increases the photoisomerization.

- b. for whether Cy3 is totally rotationally obstructed – by fluorescence anisotropy decays. If stacking interactions occur between Cy3 and the DNA 5'-terminus, then the fluorescence anisotropy should be decaying very slow, with no fast component, because the fluorophore lacks rotational freedom

Following the reviewer's suggestion, we performed fluorescence anisotropy experiments in the 3-context of the FEN1/DF system in both steady-state and time-resolved modes (Supplementary Fig. 2g,h). In the steady-state mode, we witnessed an

increase in the mean rotational correlation time upon hybrid substrate formation followed by a substantial decrease upon FEN1 binding. The increase in the mean rotational correlation time upon substrate formation does not directly support the formation of interactions between the dye and the hybrid DNA, since it also reflects the increase in the gyration radius of the hybrid DNA. However, the decrease in the mean rotational correlation time upon protein binding can only be associated with the release of a fast-rotating component, since protein binding would lead only to an increase in the gyration radius, which is expected to increase the mean rotational correlation time. With this in mind, we decided to follow the fast component of the anisotropy decay in time-resolved fluorescence anisotropy measurements. All the anisotropy decays fit well to bi-exponential decays with a slow component and fast component. The bi-exponential decays were modeled with “wobbling-in-cone” (local motion of the fluorophore) (Reference 13), with the cone attached to a cylinder (global spinning motion of the DNA-dye complex around its main axis of symmetry); please see the SI Methods section for further details. The relaxation time of the fast component associated with wobbling-in-cone was in the range of previously reported values (References 11 and 13). The fast component was present in all three DNA contexts, indicating that the rotation is not fully inhibited in any of the three DNA contexts (please refer to τ_{cone} values in Supplementary Fig. 2h). For the quantification, please refer to the below answer and revised manuscript.

I would like to note that in previous works (see Lerner & Ploetz et al., J. Phys. Chem. B, 2016), it was not the absolute theoretical value of the dye available volume that was compared to the PIFE results, but rather the ratio of theoretical dye available volumes in free dsDNA and in the presence of the bound protein. By that, these authors did take into account at least the expected steric restriction of Cy3 in BOTH states. It is important to note that these studies included solely Cy3 internally-labeling dsDNA, to minimize the addition of the dye-DNA stacking interactions. If the authors would like to be even more accurate, they can use the method developed by Börner and co-workers in the paper about NAIFE. In fact, if also specific dye-DNA interactions are involved, a better model can be developed, in which for a fraction of time, f_1 , the dye is stuck in trans isomer (an approximation) and in the remaining $1-f_1$ it is again acting as an excited-state molecular rotor where photoisomerization is inhibited by steric restriction. Although comparing dye available volumes of each case has been utilized for the rationalization of PIFE, it has not been put to the test in this work. This part is definitely missing in this work. Therefore, I request a comparison of the experimental results to the theoretical calculations of the isomerization/rotational restriction, using either one of the theoretical calculations proposed in the literature.

Both studies by Lerner & Ploetz *et al.* and Börner and co-workers explain fluorescence modulation in terms of different ratios of volumes; therefore, they construct their models with spatial consideration. With no direct structural data, the volumes involved in existing models rely mainly on theoretical calculations/computational predictions. The model developed by Lerner & Ploetz *et al.* accounts for PIFE as a proportionality between the isomerization rates and a relative change in the accessible volume of the

dye, determining a proportionality factor B, which accounts “for solvent properties, tethering of the dye to the DNA, other specific interactions, etc.”. If the nature of the interactions changes enough, as in the case of our DF system, the effect of factor B can totally change the final outcome even when opposed to the effect of the relative ratio of the accessible volumes. In the model used by Börner and co-workers, the ratio of the contact to the accessible volumes is defined as a parameter that varies continuously to produce lifetimes between the lifetime of Cy3 and Cy3B. We opted not to decouple the lifetimes into its components, since the lifetime decays of conjugated-Cy3 are already multi-exponential and complex to interpret.

We however opted to test the correlation between the isomerization and the rotational restriction using an experimentally measurable property. We therefore defined the photoisomerization rate (k_{iso}) from experimental lifetimes by considering a continuous variation of Cy3 lifetimes between free Cy3 and Cy3B as suggested by Börner and co-workers. To access the rotational restriction, we followed the wobbling-in-cone diffusion coefficient (D_w) (associated with the fast component) as previously defined (SI References 11, 13 and 14). This coefficient incorporates both steric effects through the angle of the cone, and consequently its volume, as well as the rate of rotation. We found that k_{iso} correlates well with D_w (Supplementary Figs. 2h and 3b). Since D_w also includes the rate of rotation, it accounts not only for the steric effect but also for the transient interactions that would be observed through slowing rotation, as the fluorophore would temporarily adhere to the surface of the cone. These interactions can delay the photoisomerization long enough to lead to a net increase in radiative de-excitation. Since the FEN1/DF system is complicated by the hybrid DNA, we first checked the validity of our correlation using a set of both 3' and 5'-terminally labeled oligos (Supplementary Fig. 3b). It is worth mentioning that the correlation can account for 5' stacking as both the 5' and 3'-terminally labeled oligos behaved similarly. In the FEN1/DF system, this correlation broadly applied. With this, we conclude that considerable interactions are formed in the hybrid DNA beyond 5' stacking and that FEN1 removes both these interactions and 5' stacking, leading to a slightly lower lifetime in the presence of FEN1 compared to in the presence of the oligo alone. Nevertheless, we do not distinguish between the particular types of interactions and we just refer to the overall structure as dictated by all the interactions, the fluorophore position and the DNA sequence. We simplified the terminology to appeal to the general audience of *Nature Communications* and to focus on the applicability of fluorescence modulation to biological questions.

It is not true that reduction on fluorescence was not reported in the context of PIFE. It has been used to measure the extrusion of nontemplate bases in initially transcribing complexes in bacterial transcription initiation (Ploetz & Lerner et al. Sci. Rep., 2016). It is just that the fluorescence has decreased in this stage relative to RPo stage, and this was expected if the Cy3-labeled base got extruded out of the protein surface. I request the authors to emphasize the relativeness of the method.

We do not define PIFQ as a reduction of a pre-existing PIFE but rather as a stand-alone effect starting from the DNA alone (initial state) followed by quenching upon protein binding (final state).

The mentioned study, even though showing a reduction of fluorescence, simply defines the initial state as a protein-bound PIFE state, where the PIFE decreases upon the extrusion of nontemplate bases. We define PIFQ and PIFE on the basis of fluorescence modulation upon protein binding starting from the DNA as an initial state. This is most clear in the cases of RPA/SSB/gp2.5, where the ssDNA substrates showed both fluorescence enhancement and quenching upon protein binding. The quenching is not a result of reversing a pre-existing PIFE effect. In the case of FEN1, we can argue that there is a previous enhancement due to hybrid substrate formation, but again we define quenching from the initial state of the DF substrate. The fluorescence enhancement due to substrate formation is now called NAIFE, as indicated by both Reviewers 1 and 2, in accordance with the literature. Nevertheless, PIFQ from the point of view of FEN1 binding still stands as a measurable quenching effect on its own.

Finally, I would like to mention that most importantly, Cy3 fluorescence changes, assessed from ensemble-averaged experiments, are not as accurate as should be. Such results only qualitatively teach the readers about PIFE or PIFQ, since it is hard to make sure all components were bound.

Even a report of the fluorescence lifetime components of ensemble-averaged experiments, and how these components change between different states is insufficient, since Cy3 already has a multiexponential decay.

We performed ensemble experiments at saturating protein concentrations (nearly 100 fold above their equilibrium dissociation constants). The proteins used in this study all exhibited K_d in the low nano-molar range. In FEN1 DF substrates, the labeled oligo was always used in the limiting concentration when annealing the substrates. Furthermore, the substrates were purified and validated on non-denaturing PAGE. Our findings related to FEN1 and RPA quenching were validated in single molecule assays as well. Even in the improbable case that binding would not be saturated, this would lead only to a change in the amplitude of modulation and not in the direction of it.

In my view, the best test of PIFE versus PIFQ is with the single-molecule experiments, presented in the last part of the manuscript. Additionally, in light of the alternative explanation I gave of a possible dual effect of Cy3 labeling the 5'-terminus of DNA, the only outstanding PIFQ result is that with the O-328 construct.

Based on our aforementioned arguments and our new data on FEN1/DF and ssDNA-binding proteins systems, we believe that the reviewer's alternative explanation based mainly on 5' stacking does not describe the quenching effect we witness in our systems. Moreover, in the revised manuscript, with the addition of two more ssDNA-

binding protein systems, we witness significant (30%) quenching with several oligos with different labeling positions and sequences. We also observed the quenching upon annealing of the complementary strand in a NAIFE-analogous manner, an effect that we refer to as NAIFQ.

Regarding the single-molecule experiments, I find it interesting that Cy3 fluorescence-change experiments are compared to independently acquired smFRET experiments. Why not perform both? PIFE and FRET can be combined, either using alternating laser excitation (Ploetz & Lerner et al., Sci. rep., 2016) or via immobilized assays looking both at FRET and at the sum of donor & acceptor fluorescence intensities (Craggs et al., Nucleic Acid Res., 2014). This way it can be shown unambiguously as a Cy3 intensity change and not FRET change.

We followed quenching with FRET in the same experiment, as the reviewer suggested and has been reported earlier (Craggs et al., Nucleic Acid Res., 2014). The experimental results (Fig. 5c in revised manuscript) agree with our earlier conclusion that FRET and quenching happen at different timescales and we postulate that they report on two different sequential steps in FEN1 binding processes.

Regarding the lifetime of Cy3 internally labeling O-328 that is higher than Cy3B and the claim that it is because of the rigidification formed by secondary structure formation. But still, this means that the Cy3 isomer that was stabilized is more fluorescent than Cy3B. Some other factor has influenced Cy3 to have different photophysical parameters than Cy3. Was there a spectral change associated with this finding? Can you elaborate on that?

We apologize for not being clear about the basis of the high fluorescence in O-328. Our initial manuscript included in the result section a possibility that other pathways beyond photoisomerization could also contribute to this high fluorescence. This is what was stated in the original manuscript:

“Finally, since O-328 stood out as having the highest Cy3 fluorescence, even higher than Cy3B, we aimed to further probe the basis of its high fluorescence. We postulated that such high fluorescence could be due to the rigidification of the excited trans state imposed by the overall DNA structural configuration and possibly the additional effect of other pathways beyond photoisomerization.”

However, we are addressing experimentally the reviewer’s comment in the revised manuscript. We show that there is no significant change in the absorption/emission spectra upon K⁺ binding, indicating that quantum energy levels are unperturbed in O-328/K⁺ (Supplementary Fig. 4e). We also ruled out any ground state complex formation by monitoring the extinction coefficient of O-328 in the presence and absence of K⁺ (Supplementary Fig. 4e). We therefore compared the lifetime-generating rates to those of Cy3B and showed that the high fluorescence stems from a combination of slightly prolonged radiative de-excitation and reduction in photoisomerization-independent non-radiative pathways (e.g., internal conversion) (Supplementary Fig. 4f).

Regarding the Cy3B control experiments, the authors write that

“To probe whether the fluorescence modulations in FEN1/DF system are analogous to those in PIFE, particularly with respect to the modulation of photoisomerization, we exchanged the iCy3 fluorophore with a Cy3B in our 3-context system of oligo, DF, and DF/FEN1 for flap lengths of 2, 4 and 6 nt. Cy3B is an analog of Cy3 dye, but with a rigid inter-heterocyclic construction rather than a rotationally flexible polymethine bond (Supplementary Fig. 1).” and also

“In our study, the fluorescence lifetimes of the Cy3B-bearing oligos alone and their corresponding DF substrates either with or without FEN1 showed no difference (Fig. 2e).”. Does Cy3B labeling the 5'-terminus, also stack to it as does Cy3? How much? Is it comparable to the stacking interaction of Cy3? I suggest to perform this comparison, using the same construct with Cy3B, only comparing on the basis of fluorescence anisotropy.

We thank the reviewer for this suggestion of using Cy3B-DNA since it is a simpler system for checking the effect of DNA-dye interactions without the added complication of photoisomerization pathway. Nonetheless, these findings can be cautiously extrapolated to Cy3 up to the chemical and structural differences. Following the reviewer's suggestion, in the revised manuscript, we compared the fluorescence anisotropy of Cy3 and Cy3B by labeling the 5' terminus in the oligo and DF substrate (Supplementary Fig. 2g,h). Our results suggest that both Cy3 and Cy3B interact with the oligo DNA at a similar level. However, the interactions within the hybrid DNA substrate context are reduced in the case of Cy3B.

Minor:

1. “maybe be attributed to the different lengths” should be changed to “maybe attributed to the different lengths”

We apologize for this typo and we have corrected it in the revised manuscript.

2. The authors wrote “Even though both isomers can absorb light upon excitation, only the trans S1* -to-trans S0 transition results in fluorescence emission. Furthermore, several lines of evidence support that the ground state is primarily trans 4,6-8, with some studies even considering it to be all trans 9,10”. This statement is inaccurate. The correct statement is that although both cis and trans isomers exist in ground-state, it is mostly the trans isomer that is excited and not the cis one due to large dipole strength of the trans isomer and the very low dipole strength of the cis isomer (see works from Marcia Levitus proving this using transient absorption experiments).

We have taken the reviewer's suggestion into consideration and revised the statement accordingly.

3. Regarding the Jablonski diagram in Fig. 1: if the 90-degree intermediate is tightly coupled

to the ground-state, and if the dipole strength of the cis transitions is negligible, then why at all show the transitions into, out of and within the cis excited-state? It is rarely visited.

The revised Fig. 1 does not include these transitions.

Reviewer #3 (Remarks to the Author):

In this manuscript the authors contributing to the understanding of fluorescent enhancement and quenching of labeled DNA in the presence of proteins. In particular, the authors argue that the phenomenon of protein-induced fluorescence enhancement is probably mostly a special case of a more general modulation of fluorescence that can also include quenching, which the authors term protein-induced fluorescence quenching. It is clear that both DNA and proteins can affect the rate of cis-trans photoisomerization, and hence fluorescence intensity, of many commonly used cyanine dyes. It is well known that the nucleobase sequence has a strong influence on this rate. Since the fluorescence of these dyes increases in the presence of nucleobases, it makes sense that the addition of proteins to the mix can either increase the fluorescence either by enhancing the dye DNA interaction or by an independent effect that also contributes to a greater effective local viscosity. While it may be the case that many proteins can further enhance fluorescence, it seems clear that if the DNA sequence (or structure) is already strongly enhancing, further enhancements are unlikely and more likely to be quenching.

I think this is a strong manuscript that addresses an important topic. DNA-dye-protein systems are extremely common in many experimental contexts and the framework and results in this manuscript are likely to be very helpful for many researchers in designing effective experiments and in avoiding experimental artifacts due to the high environmental sensitivity of common cyanine dyes such as cy3 and cy5.

The manuscript is very well written and presents the data clearly. The results are supported by an extensive set of appropriate and comprehensive experiments. In my opinion, the manuscript is appropriate for publication in Nature Communications as is or with minor revisions.

We thank the reviewer for the positive feedback on the manuscript.

One aspect that I found unnecessarily confusing is the non-standard usage of iCy3 and iCy5 for dyes conjugated to the DNA with phosphoramidites, and Cy3N for dyes conjugated via NHS. In particular, the naming scheme differs from the one used within the sequences in Supp. Table 1, which uses iCy3 for internal labeling. I would encourage the authors to adopt a more intuitive naming scheme such as: 5'-Cy3 instead of iCy3 (and equivalent for Cy3 or 3'-end labeling), iCy3 for internal labeling with amidite, and 5'/i/3'-NHS-Cy3/5 for the various forms of labeling via dye-NHS.

We thank the reviewer for this suggestion. In the revised manuscript, we have changed the naming scheme to simplify the different labeling chemistries and positions. We have adopted the following nomenclature in the revised text and figures: pCy3 to denote a phosphoramidite-labeled Cy3 proceeded by the position of the labeling (5', 3' or internal) and nCy3 to denote an NHS-labeled Cy3.

Another detail that seems important is a short summary of the actual sequence dependence that was observed in the fluorescence. There are quite a few tested oligos, so it might be possible to say something about the effect of the specific identity of the nucleobases immediately adjacent to the dyes.

We agree with the reviewer that the data presented in this manuscript would be suitable for such a pattern study to reveal adjacent nucleobase-specific changes in Cy3 fluorescence. It has been previously reported that adjacent nucleobases affect Cy3 fluorescence (References 18-20 main text). However, we believe that the overall structure of the dye-oligo complex is the determining factor in dictating the fluorescence properties of Cy3. This structure is difficult to predict and further studies (for example NMR) could be performed to decipher the role of each of these factors.

Reviewers' comments:

Reviewer #1 (Remarks to the Author):

The authors addressed most of my concerns.

Reviewer #2 (Remarks to the Author):

The revised manuscript “Initial state of the DNA-dye complex sets the stage for on demand fluorescence modulation upon protein binding” by Rashid, Raducanu & Zaher et al., is an interesting examination of the current rationalization behind protein-induced fluorescence enhancement (PIFE), which can be written more as induced fluorescence modulation (IFM or something like that). In my first review I had many questions and suggestions – most of them were given satisfactory answers. Overall, the revised version of the manuscript is improved. I do still have some general comments:

Regarding the request to perform transient absorption experiments: the authors decided that these experiments are out of the scope for this paper. The authors provided a thorough explanation for why their conclusions can be extrapolated from previous works who did perform the transient absorption experiments. I kindly ask the authors to add the reasoning they provided in the response also in text to the text of the manuscript. That comment is true also for any type of textual reasoning given in the response to the authors, but not in the manuscript text itself.

Regarding the fluorescence anisotropy decay measurements, the authors added: the authors wrote that judging from the existence of the fast rotating correlation time and the fact that it was present in all three DNA contexts, indicating that the rotation is not fully inhibited, does not mean there is no effect. The authors should report the values of the fast component's amplitude and report on any changes in its values. Reductions in the amplitude of the fast component of the fluorescence anisotropy decays should report on reduction in rotational freedom.

In the response to the reviewers, the authors stated they did not try to extract information from the fluorescence lifetime components, because the decays were already multi-exponential and complex to interpret. I would like to mention that this fact is known and has already been treated theoretically as such. For instance, in Lerner & Plötz et al., a model was developed to treat the Cy3 excited-state behavior. Therefore, the fact that “lifetime decays of conjugated-Cy3 are already multi-exponential and complex to interpret”, as the authors stated, cannot serve as a reason not to analyze them properly. In this context, we would like to mention that if authors of an existing work choose not to use prior developments, it is still customary to cite scientific works prior to the current work, especially if the current work serves as an advancement of the existing knowledge relative to the existing previous publication.

Eitan Lerner

Reviewers' comments:

Reviewer #1 (Remarks to the Author):

The authors addressed most of my concerns.

We would like to thank the reviewer for the thoughtful comments and efforts towards improving our manuscript.

Reviewer #2 (Remarks to the Author):

The revised manuscript “Initial state of the DNA-dye complex sets the stage for on demand fluorescence modulation upon protein binding” by Rashid, Raducanu & Zaher et al., is an interesting examination of the current rationalization behind protein-induced fluorescence enhancement (PIFE), which can be written more as induced fluorescence modulation (IFM or something like that). In my first review I had many questions and suggestions – most of them were given satisfactory answers. Overall, the revised version of the manuscript is improved.

We thank the reviewer for the insightful remarks during both review stages that have led to the improvement in both the quality and the presentation of this manuscript.

I do still have some general comments:

Regarding the request to perform transient absorption experiments: the authors decided that these experiments are out of the scope for this paper. The authors provided a thorough explanation for why their conclusions can be extrapolated from previous works who did perform the transient absorption experiments. I kindly ask the authors to add the reasoning they provided in the response also in text to the text of the manuscript. That comment is true also for any type of textual reasoning given in the response to the authors, but not in the manuscript text itself.

As requested by the reviewer, we have now added two paragraphs in the discussion explaining our reasoning for not performing the transient absorption experiments as well as discussing the previous theoretical models.

Regarding the fluorescence anisotropy decay measurements, the authors added: the authors wrote that judging from the existence of the fast rotating correlation time and the fact that it was present in all three DNA contexts, indicating that the rotation is not fully inhibited, does not mean there is no effect. The authors should report the values of the fast component's amplitude and report on any changes in its values. Reductions in the amplitude of the fast component of the fluorescence anisotropy decays should report on reduction in rotational freedom.

We agree with the reviewer that the existence of fast-rotating correlation time in all three DNA contexts does not mean there is no effect. In fact, our conclusion regarding the

time-resolved anisotropy stated that ***“there are interactions restricting the dye’s rotational freedom in the three-context system and these interactions increase upon substrate annealing and decrease upon FEN1 binding.”*** We would like to comment that this statement is not based on the changes of the fast-rotating correlation time alone but rather on the composite parameter, Wobbling Diffusivity (D_w), which incorporates both the fast-rotating correlation time and the fractional amplitude of the fast component, as described in detail in SI Methods. During the fluorescence emission lifetime, as the reviewer previously stated: ***“for a fraction of time, f_1 , the dye is in trans isomer (an approximation) and in the remaining $1 - f_1$ it is again acting as an excited-state molecular rotor where photoisomerization is inhibited by steric restriction,”*** the fractional amplitude of the fast component would report on the fraction of time (or by ergodic equivalence, the fraction of molecules) for which the fluorophore is free to rotate during the lifetime of the excited state, regardless of the rate of rotation. We believe that this rate of rotation is equally important to assess the rotational freedom.

It is noteworthy that we did not use a simple sum of two-exponential decay framework, but a model that uses parameters with physical attributions. In the case of a model based on the sum of two exponentials, we agree with the reviewer that the fractional amplitude will directly report on the rotational freedom. We do not believe that this simple model can describe the anisotropy decays of a general DNA-Dye system due to the existence of interference between dye-wobbling and DNA-spinning motions, and in some cases, the existence of considerable residual anisotropy. Moreover, by comparing the expression of D_w with the Taylor expansion of the fast component decay, it can be observed that in the first order, this parameter accounts for the initial slope of the anisotropy decay. Hence, unlike the fractional amplitude, D_w has a physical meaning, which is robust to fitting methods and mathematically incorporates both the fractional amplitude of the fast component as well as the fast-rotating correlation time.

In the response to the reviewers, the authors stated they did not try to extract information from the fluorescence lifetime components, because the decays were already multi-exponential and complex to interpret. I would like to mention that this fact is known and has already been treated theoretically as such. For instance, in Lerner & Plötz et al., a model was developed to treat the Cy3 excited-state behavior. Therefore, the fact that “lifetime decays of conjugated-Cy3 are already multi-exponential and complex to interpret”, as the authors stated, cannot serve as a reason not to analyze them properly. In this context, we would like to mention that if authors of an existing work choose not to use prior developments, it is still customary to cite scientific works prior to the current work, especially if the current work serves as an advancement of the existing knowledge relative to the existing previous publication.

We apologize for missing citation by Lerner & Plötz et al. We have also cited the relevant papers in the revised manuscript. We appreciate both theoretical models and believe that they can be extrapolated to all induced fluorescence modulations that were observed in this work. However, for both models to be applicable to quenching, one should consider the possibility that the dye in the initial state can be more restricted than in the final state

due to various interactions and DNA overall structures. Both models are now briefly discussed in the revised manuscript with appropriate citations. We would like to emphasize that throughout the paper, we chose not to delve into the theoretical framework, not because the existing work is inapplicable or too complex, but rather to derive our conclusions from simple physical observables. This is in line with our aim to keep the general practical appeal of our findings to broader audience. In this context, our conclusions are sufficiently supported by the average lifetime without the need to assign an interpretation of the lifetime components.

REVIEWERS' COMMENTS:

Reviewer #2 (Remarks to the Author):

The state of the manuscript is now very good, in my opinion. I have no further questions or comments.

Eitan Lerner